# Ensemble-based estimation of sea-ice volume variations in the Baffin Bay

Chao Min[1,2], Qinghua Yang[1,2], Longjiang Mu[3], Frank Kauker[3,4], Robert Ricker[3]

[1]School of Atmospheric Sciences, Sun Yat-sen University, and Southern Marine Science and Engineering Guangdong Laboratory (Zhuhai), Zhuhai, 519082, China

[2]State Key Laboratory of Numerical Modeling for Atmospheric Sciences and Geophysical Fluid Dynamics, Institute of Atmospheric Physics, Chinese Academy of Sciences, Beijing, 100029, China

[3]Alfred Wegener Institute, Helmholtz Centre for Polar and Marine Research, 27570 Bremerhaven, Germany

[4]Ocean Atmosphere Systems, 20249 Hamburg, Germany

*Correspondence to*: Qinghua Yang (yangqh25@mail.sysu.edu.cn)

**Abstract.** Sea ice in the Baffin Bay plays an important role in deep water formation in the Labrador Sea and contributes to the variation of the Atlantic meridional overturning circulation (AMOC) on larger scales. Sea-ice data from locally merged satellite observations (Sat-merged SIT) in the Eastern Canadian Arctic and three state-of-the-art sea ice-ocean models are used to quantify sea-ice volume variations from 2011 to 2016. Ensemble-based sea-ice volume (SIV) fluxes and the related standard deviations in the Baffin Bay are generated from four different estimates of SIV fluxes that derived from Sat-merged SIT, three modeled SITs and satellite-based ice-drift data. Results show that the net increase of the SIV in Baffin Bay occurs from October to early April with the largest SIV increase in December ($113\pm17$ km$^3$ month$^{-1}$) followed by a reduction from May to September with the largest SIV decline in July ($-160\pm32$ km$^3$ month$^{-1}$). The maximum SIV inflow occurs in winter with the amount of $236(\pm38)$ km$^3$ while ice outflow reaches the maximum in spring with a mean value of $168(\pm46)$ km$^3$. The ensemble mean SIV inflow reaches its maximum ($294\pm59$ km$^3$) in winter 2013 caused by high ice velocity along the north gate while the largest SIV outflow ($229\pm67$ km$^3$) occurs in spring of 2014 due to the high ice velocity and thick ice along the south gate. The long-term annual mean ice volume inflow and outflow are $411(\pm74)$ km$^3$ and $312(\pm80)$ km$^3$ year$^{-1}$, respectively. Our analysis also reveals that on average, sea ice in the Baffin Bay melts from May to September with a net reduction of 335 km$^3$ in volume while it freezes from October to April with a net increase of 218 km$^3$. In the melting season, there is about 268 km$^3$ freshwater produced by local melting of sea ice in the Baffin Bay. In the annual mean, the mean freshwater converted from SIV outflow that enters the Labrador Sea is about 250 km$^3$ year$^{-1}$ (i.e., 8 mSv), while it is only about 9% of the net liquid freshwater flux through the Davis Strait. The maximum freshwater flux derived from SIV outflow peaks in March with the amount of 65 km$^3$ (i.e., 25 mSv).

# 1 Introduction

Baffin Bay is a semi-enclosed basin between Ellesmere Island, Baffin Island and Greenland. This bay serves as an important pathway of southward flowing and cold freshwater draining off from the Arctic into the North Atlantic Oceans (Curry et al., 2010; Curry et al., 2014). Freshwater outflows through Davis Strait entering the Labrador Sea are integrated from Canadian Arctic Archipelago and west Greenland glacial runoff, river inputs, sea ice meltwater and precipitation (Curry et al., 2010; Curry et al., 2014; Tang et al., 2004). Locally, sea ice in Baffin Bay has a significant influence on Greenland coastal air

temperatures and ice sheet surface-melt (Ballinger et al., 2018; Rennermalm et al., 2009; Stroeve et al., 2017). The sea ice condition in Baffin Bay also impacts wildlife habitations (Ferguson et al., 2000; Laidre and Heide-Jørgensen, 2004; Spencer et al., 2014). Furthermore, marine-based activities, such as shipping, are strongly influenced by the sea ice conditions in the bay (Pizzolato et al., 2016). Therefore, understanding the sea ice variations in the Baffin Bay is of strong interest for climate change research but also for stakeholders.

Landy et al. (2017) composed a 14-year SIT data set in the eastern Canadian Arctic (ECA) from ICESat, CryoSat-2 and passive microwave (PMW) snow depths, then merged with SMOS where the mean CryoSat-2 thickness is <1 m. This Sat-merged SIT data are utilized to calculate the local sea ice volume variation in the Baffin Bay but not the sea ice volume fluxes and thermodynamic growth (Landy et al., 2017). However, seasonal thin sea ice in the bay is dominating and satellite-based ice thickness has large errors in the bay with respect to other regions in the Arctic Basin. For example, SMOS SIT usually

underestimates the ice thickness when the ice is thicker than 1.0 m and CryoSat-2 SIT has large uncertainties for thin ice below 1.0 m (Ricker et al., 2014; Tian-Kunze et al., 2014; Tietsche et al., 2018). In a recent study, Bi et al. (2019) analysed the sea-ice area fluxes in Baffin Bay on a long-term time period and the increasing trend of the annual sea-ice area flux are found to be $38.9 \times 10^3$ km$^2$ decade$^{-1}$ for the inflow through the north gate, $7.5 \times 10^3$ km$^2$ decade$^{-1}$ for the inflow through Lancaster Sound and $82.2 \times 10^3$ km$^2$ decade$^{-1}$ for the outflow through the south gate (Davis Strait), respectively. However, sea-ice

volume variations in Baffin Bay, strongly controlled by sea-ice volume inflow and outflow, are not investigated in that study. Cuny et al. (2005), Tang et al. (2004) and Kwok (2007) estimated the annual mean SIV outflow through Davis Strait into the Labrador Sea based on simple assumptions of linear variation of mean SIT across the strait due to scarce SIT observations. They reported mean SIV outflows through Davis Strait of about 528 km$^3$ year$^{-1}$, 873 km$^3$ year$^{-1}$ and 530-800 km$^3$ year$^{-1}$, respectively. Until several years ago, the mean SIV outflow (407 km$^3$ year$^{-1}$, from 2004 to 2010) averaged from November to

May is approximately presented with the SIT observations from five upward looking sonars (ULSs) that are moored in the Davis Strait rather than a simple SIT assumption (Curry et al., 2014). However, to the authors' knowledge, there is no study investigating the year-round SIV inflow and outflow covering the years of the lowest sea-ice extent records (i.e., 2012 and 2016). The freshwater budget is a function of sea-ice formation and melting, input from river water and land-ice input (Landy et al., 2014; Landy et al., 2017). The sea-ice thermodynamic processes are closely related to the desalination of seawater and

the freshwater budget in the Baffin Bay. For instance, during sea-ice freezing, salt is discharged into the surface ocean water leading to denser and saltier conditions which destabilizes the water column. On the other hand, when the sea ice melts

fresh/hyposaline water is drained into the surface water causing desalination of the surface water and, consequently, stabilizes the water column.

In this study, we focus on the local sea-ice volume variations in Baffin Bay. We define the SIV inflow and outflow gates following Kwok (2007) to be located at ~73°N and ~68°N between Baffin Island and Greenland (Fig. 1), respectively. The sea-ice imported into Baffin Bay through the north gate can be divided into three sources: Sea ice input (including multi-year ice) from Nares Strait, Lancaster Sound and Jones Sound that originates from the Arctic Ocean and the Canadian Arctic Archipelago (CAA) and a large amount of ice generated in polynyas, i.e., the North Water (NOW) Polynya (Bi et al., 2019; Kwok, 2005, 2007). In our study, we focus on the total amount of sea-ice inflows through the north gate summing up the ice from the Arctic Ocean, the CAA and the NOW Polynya. Sea-ice volume variations are calculated in the area between the north gate and the south gate. There is limited *in-situ* observed SIT in this bay. Also, SMOS, Cryosat-2 and CS2SMOS have large uncertainties in that area (Ricker et al., 2014; Ricker et al., 2017; Tian-Kunze et al., 2014; Tietsche et al., 2018). For instance, SMOS SIT is underestimated because (1) SMOS only provides valid SIT for ice thinner than 1 m and (2) the 100% ice concentration assumption during the data retrieval is not fulfilled (Tian-Kunze et al., 2014; Tietsche et al., 2018). To address the challenging estimation of sea-ice volume variations in Baffin Bay, locally merged satellite SIT data (Landy et al. 2017, 2019, 2020) and three sea ice-ocean models driven by atmospheric reanalysis are employed, namely the sufficiently well validated combined model and satellite sea ice thickness (CMST), the widely used Pan-Arctic Ice Ocean Modeling and Assimilation System (PIOMAS), a version of the North Atlantic/Arctic Ocean Sea Ice Model (NAOSIM) with optimized parameters. Because very little *in-situ* observations can be used for validation in Baffin Bay, we carry out an inter-comparison between CMST, NAOSIM, PIOMAS, the Towards an Operational Prediction system of the North Atlantic and European coastal Zones (TOPAZ4) and the merged satellite SIT of Landy et al. (named Sat-merged SIT hereafter). To obtain an estimate of the sea-ice volume fluxes, we calculate the ensemble mean of the inflows and outflows from the three modeled SITs, Sat-merged SIT and satellite-based ice drift. Furthermore, since the Baffin Bay plays a crucial role as the primary source of freshwater and sea ice in the Labrador Sea (Curry et al., 2014; Tang et al., 2004) the amount of freshwater flux exported into the Labrador Sea is calculated based on the estimated outflowing SIV through the Davis Strait.

This paper is organized as follows: Sea ice data sets and computing methods used in this study are described in section 2. In section 3, we present the major findings. Discussions of SIV flux uncertainties and freshwater fluxes are given in section 4. In section 5, main findings are finally drawn.

## 2 Data and methods

### 2.1 CMST sea ice data

The complementary of SMOS SIT and CryoSat-2 SIT is utilized in CMST by assimilating SMOS SIT from University of Hamburg, CryoSat-2 SIT from AWI and Special Sensor Microwave Imager/Sounder (SSMIS) ice concentration processed at IFREMER into the MITgcm (Mu et al., 2018a). The sea ice-ocean model is forced by ensemble atmospheric forecasts from

the UK Met Office (UKMO) taking the uncertainty of the atmospheric data into account (Yang et al., 2015). CMST provides daily sea-ice thickness (SIT), concentration (SIC) and drift (SID). CMST SIT was systematically validated within the Arctic basin by Mu et al. (2018a) and its SID were further validated against NSIDC and SAR data in the Fram Strait by Min et al. (2019). Additionally, CMST is successfully applied to obtain a relatively accurate estimation of the year-round sea ice volume export through the Fram Strait (Min et al., 2019).

## 2.2 NAOSIM sea ice data

The NAOSIM SIT data are produced by a regional sea ice-ocean model of the Arctic and northern North Atlantic Ocean (NAOSIM) developed at the Alfred Wegener Institute (Köberle and Gerdes, 2003; Kauker et al., 2003; Karcher et al., 2007). The model is forced by the NCEP Climate Forecast System version 2 (Saha et al. 2014). 15 model parameters (e.g., ice strength, drag coefficients) were optimized simultaneously using a micro genetic algorithm (mGA). A detailed description of NAOSIM and the methodology used for the optimization can be found in Sumata et al. (2019a, b). The model version used in this study distinguishes from the model version applied for the optimization in Sumata et al. (2019a, b) by a horizontal resolution of about 28 km (Model version MR in Sumata et al. (2019a)). The parameters (except the vertical mixing coefficient) are taken from the third optimization of Sumata et al. (2019b) termed OPT-3.

## 2.3 PIOMAS sea ice data

The widely used Pan-Arctic Ice-Ocean Modeling and Assimilation System (PIOMAS) SIT data are produced by a sea-ice ocean model that assimilates near-real-time daily SIC from National Snow and Ice Data Center (NSIDC) and sea surface temperature in the ice-free areas from the National Center for Environmental Prediction (NCEP) and National Center for Atmospheric Research (NCAR) reanalysis by nudging and optimal interpolation (Schweiger et al., 2011; Zhang and Rothrock, 2003). It is forced by atmospheric data from the NCEP/NCAR reanalysis (Schweiger et al., 2011; Zhang and Rothrock, 2003). Effective sea-ice thickness data are provided operationally from 1978 on and is permanently updated. In this study, we use the monthly SIT data of PIOMAS V2.1 from 2011 to 2016.

## 2.4 TOPAZ4 sea ice data

TOPAZ4 is a regional ocean and sea-ice prediction system. The ocean model is based on the Hybrid Coordinate Ocean Model (HYCOM version 2.2) (Bleck, 2002; Chassignet et al., 2003). The sea-ice model employs the one-thickness category and elastic-viscous-plastic rheology (Bouillon et al., 2013; Hunke and Dukowicz, 1997). This system is forced by ERA-interim atmospheric reanalysis. Ocean and sea ice observations are assimilated into TOPAZ4 (e.g., the along track sea level anomaly and gridded sea surface temperature, OSI-SAF sea ice concentration and drift, and CS2SMOS SIT) (Xie et al., 2018). Since the TOPAZ4 reanalysis data cover a short period from 2014 to 2018, the TOPAZ4 SIT and SID are only used for inter-comparison with the other sea ice data but not for any volume or flux calculations in this study.

**2.5 Sat-merged SIT data**

Because in-situ observations of SIT are very scarce in Baffin Bay, a locally merged satellite SIT (Sat-merged SIT) data set is utilized to calculate the SIV variations during the freezing season since this data set are used to estimate the sea ice variations in the Eastern Canadian Arctic including Baffin Bay before. This Sat-merged SIT data are calculated from CryoSat-2 radar freeboards (accessed from the European Space Agency) and passive microwave (PMW) snow depths (available from NSIDC at https://nsidc.org/data/NSIDC-0032/versions/2) and then merged with SMOS SIT (available from the University of Hamburg

at https://icdc.cen.uni-hamburg.de/en/l3c-smos-sit.html) where the mean CryoSat-2 thickness is <1 m. More details about this data set can be found in Landy et al. (2017, 2019, 2020).

**2.6 NSIDC SID data**

The Polar Pathfinder Daily 25 km EASE-Grid sea ice drift data (V4) from NSIDC are used to calculate SIV fluxes because it contains year-round data for the time period investigated. The AVHRR, AMSR-E, SMMR, SSM/I, SSMIS, International

Arctic Buoy Program (IABP) buoys observations and reanalysis wind data are integrated to derive the NSIDC sea ice motion (Tschudi et al., 2019; Tschudi et al., 2020). The NSIDC data set has been recently validated with high-resolution Envisat wide-swath SAR observations and IABP buoy measurements by Bi et al. (2019). Comparing with the observed sea ice drift that retrieved from high-resolution (~100 m) Envisat Synthetic Aperture Radar (SAR) observations, the NSIDC drift slightly underestimates the ice drift with a mean bias of -0.68 km day[-1], while it has a high correlation (R=0.87) with SAR drift (Bi et

al., 2019). The NSIDC drift data (V4) are chosen as a reference to evaluate model ice drift and are applied to calculate the sea-ice flux.

**2.7 Retrieving methods for SIV flux**

We use monthly mean sea-ice thickness and drift to obtain the SIV fluxes following Ricker et al. (2018). The formulas to derive the SIV inflows and outflows are the same as applied in Min et al. (2019):

$$Q_{flux} = L\,H\,v, \tag{1}$$

where $Q_{flux}$ represents the SIV fluxes at the north and south gates. L and H are zonally interpolated grid width and corresponding SIT along the two gates, respectively. The meridional velocity *v* is utilized to estimate the sea ice flux (inflows and outflows). The SIC is not involved in equations (1), because they are already used to calculate the effective thickness in CMST, NAOSIM and PIOMAS. It is difficult to identify the most accurate SIT simulation and ice flux estimate, so we adopt

the ensemble approach to estimate the sea-ice variations in the Baffin Bay, i.e., ensemble mean inflows and outflows are from (1) CMST SIT and NSIDC SID, (2) NAOSIM SIT and NSIDC SID, (3) PIOMAS SIT and NSIDC SID, and (4) Sat-merged SIT and NSIDC SID (equation (1)). And we use one standard deviation (i.e., +/-number) among these ensemble members (i.e., SIV fluxes estimated from different sea ice thickness data sets and NSIDC SID) to show the uncertainties of flux estimates in

this study. Analogously, the sea-ice volume in the Baffin Bay is calculated from the ensemble mean of the Sat-merged SIT,
CMST, NAOSIM and PIOMAS SIT.

Following Ricker et al. (2018), the sea ice volume variation can be derived as follows:

$$\frac{dV}{dt}=Q_{net}+(\frac{dV_{therm}}{dt}+\frac{dV_{resid}}{dt}),\qquad(2)$$

where dV/dt represents the monthly SIV change in the Baffin Bay. $Q_{net}$ is the monthly net SIV flux ($\Delta$flux) estimated by the difference between inflow and outflow. As suggested by Ricker et al. (2018) quantifying thermodynamic growth (dVtherm/dt) and residual contributions (dVresid/dt) due to dynamics and deformation is challenging. Therefore, we only consider their integral contribution. Eventually, the integral contribution of dVtherm/dt and dVresid/dt is regarded as thermodynamic SIV growth rate in this study. To distinguish ice melting and freezing, we use negative thermodynamic SIV growth rates to represent reduction through ice melting and positive rates to denote growth due to freezing.

## 3 Results

The spatial distributions of the ensemble mean SIC, SIT and SID in March, July and October are shown in Fig. 1. We have chosen these months as they typically represent the seasonal cycle. As found by Meier et al. (2006), the maximum extent occurs in March while July is the last month when sea ice is still left and the ice freeze-up starts in October. Furthermore, we present the spatial distribution of SIT especially in July when satellite-based SIT is not available due to melting processes. The ensemble mean SIT shows that the thicker ice (>1.2 m) is located east of Baffin Island in March while largest ice velocities are found near the south gate. The spatial distribution of ensemble mean SIT in March is similar to that found by Landy et al. (2017). In July sea ice thicker than 0.3 m is located near the eastern coast of Baffin Island. Focusing on the freeze-up period (October), we found ice located near the Nares Strait mostly being thicker than 0.5 m. Highest ice velocity (more than 10 km day$^{-1}$) is found near Smith Sound and Lancaster Sound by CMST (figure not shown).

The comparisons of SIT (averaged along the north and south gates) between CMST, NAOSIM, PIOMAS, TOPAZ4 and Sat-merged SIT are shown in Fig. 2a and 2b, respectively. The SIDs from CMST, NAOSIM, PIOMAS, TOPAZ4 and NSIDC SID are compared with each other as well (Fig. 2c and 2d). The SIC variation is not shown here because the models (except NAOSIM) have already taken SIC into account via the assimilation. In general, these sea ice properties show a significant annual cycle with the mean SIT thinner than 1 m for both the north and the south gates. Compared with the Sat-merged SIT, all simulations present thicker ice than Sat-merged SIT (Fig. 2a and 2b). The mean SIT averaged along the north gate is 0.72 m for CMST, 0.83 m for NAOSIM, 0.84 m for PIOMAS and 0.55 m for TOPAZ4 during the freezing season while the mean SIT is 0.56 m for Sat-merged SIT. Likewise, the mean SIT averaged along the south gate is only 0.40 m for Sat-merged SIT while the mean SITs of CMST, NAOSIM, PIOMAS and TOPAZ4 are 0.52 m, 0.61 m, 0.72 m, and 0.44 m, respectively. In general, the simulations of NAOSIM and PIOMAS show thicker sea ice than the simulations of CMST and TOPAZ4 data who assimilate satellite-observed SIT. The SIT cycles of CMST and TOPAZ4 are more consistent with Sat-merged SIT as well.

Furthermore, sea ice drift (SID) is an important contributor for sea ice flux variation on its monthly scale (Min et al., 2019; Ricker et al., 2018). For this reason, an accurate simulation of SID is another vital factor to derive sea-ice volume flux. Again, because of the all-year round coverage and the recent validation of NSIDC drift in the Baffin Bay by Bi et al. (2019), we apply NSIDC drift to calculate the sea ice flux in this study. In addition, we conduct an inter-comparison of SID between NSIDC SID, CMST, NAOSIM, PIOMAS and TOPAZ4 SID in Fig. 2c and 2d to examine the performance of these modeled SID data.

Note that the TOPAZ4 values are from 2014-2016 for the overlapping period. A fairly similar cycle of SID is shown by CMST, TOPAZ4 and NSIDC SID. However, both CMST and TOPAZ4 present higher ice velocity than NSIDC SID while NAOSIM and PIOMAS underestimate the monthly mean ice drift. Moreover, TOPAZ4 simulates the fastest ice velocity among five data sets while PIOMAS shows the lowest ice drift across the north gate. We calculate the correlation coefficients (CCs) between these model simulations and the reference NSIDC SID. The highest significant ($\alpha$=0.05) CCs (0.94 and 0.92) are found

between TOPAZ4 and NSIDC SID while it overestimates the ice drift compared to NSIDC SID by around 52% and 82% along the north gate and south gates, respectively. Also CMST shows high CCs compared with NSIDC SID in both two gates; the correlations are 0.90 (significant) along the north gate and 0.91 (significant) along the south gate with an overestimation of 40% and 70%, respectively. The ice drift produced by NAOSIM and PIOMAS show relatively low CCs against NSIDC SID. As an example, the CCs between NAOSIM and NSIDC SID drift are 0.61 (non-significant) and 0.61 (non-significant) along

00    the north and south gates, respectively. The coefficients between PIOMAS and NSIDC SID are also relatively low as it is only 0.60 (significant) for the north gate and 0.71 (non-significant) for the south gate, respectively. Although CMST and NSIDC SID correlate very well over the time span from 2011 to 2016, this modeled SID shows a large overestimation of ice drift. Therefore, we conclude that modeled SID shows large uncertainties and we calculate ice flux estimates from CMST, NAOSIM, PIOMAS and a Sat-merged SIT and NSIDC SID, i.e. without using of any modeled ice drift.

05    The monthly and seasonal mean ice inflows and outflows from 2011 to 2016 are shown in Fig. 3 and 4, respectively. The sea-ice volume (SIV) fluxes calculated by the four members show a relatively good consistency over the years considered (Fig. 3 and 4). The ensemble mean SIV inflow and outflow are 411($\pm$74) km$^3$ and 312($\pm$80) km$^3$ per year, respectively. Even though there are some discrepancies between these four fluxes calculated from the different models and Sat-merged SIT, the fluxes show a consistent cycle of seasonal variation (in term of the ensemble standard deviation). In general, the maximum of

10    ensemble mean ice inflows occur in February and March (82$\pm$12 km$^3$ month$^{-1}$ and 82$\pm$16 km$^3$ month$^{-1}$, respectively), and the ice outflow reaches its maximum in March with an ensemble mean flux of 80$\pm$21 km$^3$ month$^{-1}$. Here, we define spring as the time span from March to May, summer from June to August, autumn from September to November, and winter from December to February. Seasonal sea-ice inflows and outflows from the three models show better consistency in the inflows than outflows, which we attribute to the larger discrepancies of the ice thickness along the south gate between CMST, PIOMAS and NAOSIM.

15    On average, the maximum of ice inflow occurs in winter with a mean value of 236($\pm$38) km$^3$ while usually the ice outflow reaches the maximum in spring with a mean value of 168 ($\pm$46) km$^3$. Looking into specific years, the maximum of SIV inflow (294$\pm$59 km$^3$) occurs in winter 2013 because of the largest sea ice drift although the ice thickness is not at its maximum. The SIV inflow in the melting season (May-September) is only 9% of that in the freezing season (October–April) and the SIV

outflow in the melting season only accounts for 11% of that in the freezing season. Furthermore, to quantify the freshwater imported into the Labrador Sea, an important area of deep water formation, we convert the SIV fluxes to the freshwater fluxes according to Spreen et al. (2020):

$$(1-\frac{S_{ice}}{S_{ref}})\,(\frac{\rho_{ice}}{\rho_{water}}) \approx 0.8 \qquad (3)$$

where the sea ice salinity ($S_{ice}$) is assumed to be 4 psu, the reference seawater salinity $S_{ref}$ is 34.8 psu, sea ice density ($\rho_{ice}$) is 901.3 kg m$^{-3}$ and seawater density ($\rho_{water}$) is 1023.9 kg m$^{-3}$ (Haine et al., 2015; Serreze et al., 2006). The monthly mean freshwater fluxes are shown in Table 1. The annual mean amount of freshwater flux that exported into the Labrador Sea derived from SIV flux is about 250 km$^3$ year $^{-1}$ (i.e., 8 mSv). Relatively large freshwater fluxes are found from February to April peaking at 65 km$^3$ month$^{-1}$ (i.e., 25 mSv) in March. The annual mean freshwater directly derived from ice meltwater in previous studies is in a range from 10 mSv (i.e., 331 km$^3$ yea$^{r-1}$ of SIV; Curry et al., 2014) to 21.3 mSv (i.e., 873 km$^3$ year$^{-1}$ of SIV; Tang et al., 2004) which is larger than our estimation.

It is essential to quantify the sea ice volume variations in the Baffin Bay because the desalination of seawater and the freshwater budget are affected by the sea ice thermodynamic processes. In this study, the locally thermodynamic processes are further investigated by considering of sea ice freezing, melting and volume fluxes (Fig. 5). The ensemble mean SIV in the Baffin Bay increases from October to early April with a maximum rate of 113±17 km$^3$ month$^{-1}$ in December. It decreases from May to September with a maximum reduction rate of -160±32 km$^3$ month$^{-1}$ in July. The net ice volume flux exported into the Baffin Bay occurs from October to March with a maximum of 46±7 km$^3$ month$^{-1}$ in December. Moreover, we analyze the thermodynamic SIV growth rate that is divided into net ice freezing and melting growth in Fig. 5b. On average, we find that the ice freezes from October to April with a mean ice freezing rate of 31 km$^3$ month$^{-1}$ while the maximum freezing rate occurs in December (67 km$^3$ month$^{-1}$). The ice melting occurs from May to September with a monthly mean of -67 km$^3$ month$^{-1}$ while the maximum occurs in July (-160 km$^3$ month$^{-1}$). Taking these thermodynamic SIV growth into account, we could infer that the surface seawater salinity increases from October to April and decreases from May to September with respect to the close connection between sea ice formation/melting and the freshwater budget.

## 4 Discussions

The sea ice flowing into the Baffin Bay through the north gate is mainly from Nares Strait, Lancaster Sound, Jones Sound, and recurring polynyas, i.e., the North Water (NOW) Polynya (Bi et al, 2019; Kwok, 2007, 2005). Kwok (2005 and 2007) pointed out that the SIV export from the Arctic through the Robeson Channel becomes most active after July. We notice that the ice thicker than 0.5 m is mostly located near the Nares Strait in October companying with higher ice velocity (more than 10 km day$^{-1}$) identified near the Smith Sound and Lancaster Sound by CMST (figure not shown). We thus speculate that most of the thick ice may be exported from the Arctic since the higher ice velocity is also found in the corresponding area of the thick ice located (i.e., Nares Strait), and the faster ice is usually deemed to be a proxy for higher ice flux, which is also noticed in

previous studies (Kwok, 2005, 2007). Moreover, the sea ice motion which greatly affects the SIV fluxes may be affected by the large-scale atmospheric circulation, such as NAO and AO. So we investigated the correlation coefficients (CCs) between NAO/AO (http://www.cpc.ncep.noaa.gov, last access: 01 October 2020) and SIV inflow and outflow for the seasonal cycle (shown in Fig. 7). The CCs between NAO index and SIV inflow and outflow are 0.68 and 0.56, respectively. For AO index and SIV inflow and outflow, the CCs are 0.34 and 0.42, respectively. However, long-term (climatic) time series of NAO/AO

and sea ice fluxes are certainly required to obtain reliable linkages.

Sea ice freezing and melting processes in Baffin Bay and SIV fluxes exported through the Davis Strait are important for the deep water formation in the Labrador Sea. The annual mean sea ice growth rate in our study is 52 km$^3$ month$^{-1}$ while it is about 87 km$^3$ month$^{-1}$ estimated in a previous study (Table 3, Landy et al., 2017). Also, the monthly mean SIV variability in our study is smaller than that of Landy et al. (2017) which can be attributed to a different area of the study regions. We also notice

that the maximum of the SIV occurs in March or early April and that the period nearly coincides with the sea ice extent evolution reported by Meier et al. (2006) who found a maximum in March. We converted the monthly mean sea-ice inflow and outflow as well as the net flux and the ice growth/melting into the freshwater volume fluxes (Fig. 8). It should be noted that the meltwater (from ice melting in the bay) released into Baffin Bay reached its maximum of 156 km$^3$ month$^{-1}$ (i.e., 59 mSv) in July of 2015 while the maximal rate of sea-ice production happened in January of 2015 leading about 65 km$^3$

freshwater stored in sea ice. The maximum amount of freshwater stored in sea ice in Baffin Bay is about 240 km$^3$ in March/April. However, it is estimated by Landy et al. (2017) to be maximal in April (445 km$^3$). Because the area of our defined region is only about half of that in Landy et al. (2017), the smaller estimated freshwater storage may mostly attribute to the smaller study area. The maxima of freshwater inflow and outflow take place in the period of January to March and February to April, respectively. The maximum net freshwater flux entering the Baffin Bay is about 53 km$^3$ month$^{-1}$ (i.e., 20 mSv) in

December of 2014 while the maximum of freshwater flux derived from ice inflow and outflow are about 99 km$^3$ month$^{-1}$ (i.e., 38 mSv) in February of 2014 and 89 km$^3$ month$^{-1}$ (i.e., 34 mSv) in March of 2015, respectively. The annual freshwater flux through the Davis Strait ranges from 172 km$^3$ (i.e., 5 mSv) in 2016 to 326 km$^3$ (i.e., 10 mSv) in 2015. Annually, the mean freshwater flux derived from SIV outflow is about 250 km$^3$ year$^{-1}$ (i.e., 8 mSv) which is about 9% of the net liquid freshwater flux (93 mSv, Curry et al., 2014) through the Davis Strait. Moreover, the mean freshwater flux estimated in this study is slightly

smaller than the estimation based on ULS SIT observations (10 mSv; Curry et al., 2014). The small difference in the estimates indicates that our ensemble-based SIV fluxes seem to be reasonable and provide a novel approach to estimate the long-term SIV variation in Baffin Bay, an area with scarce SIT *in-situ* observations.

Because of the very limited *in-situ* SIT observations in the Baffin Bay, it is not possible to identify very accurately sea-ice volume and fluxes in this area. The aim of this study is to give a state-of-the-art ensemble mean estimation of SIV flux based

on a combination of model results and observations, and to conduct a first estimate of the thermodynamic growth of sea-ice volume. Additionally, this is the first study using the Sat-merged SIT and three different model outputs to estimate sea-ice variations in the Baffin Bay. We may underestimate the ice fluxes in this bay by using the NSIDC drift pointing to the fact that long-term and high-resolution sea-ice drift data in the bay still needs to be further developed. We also notice that there are

some discrepancies among Sat-merged SIT, CMST, PIOMAS and NAOSIM thicknesses. For instance, the sea ice reduction period of NAOSIM and PIOMAS start later than that of Sat-merged SIT and CMST in Baffin Bay (Fig. 6) which might be connected to the assimilation of CryoSat-2 and SMOS thickness observations in CMST while PIOMAS and NAOSIM do not. CMST SIT also shows a much more coherent ice thickness to the satellite observations, e.g., the sea ice volume variation shown by CMST reaches its maximum in March (Fig. 6) which is also found by Landy et al. (2017). However, the monthly mean variability shows a consistent start (October) of ice volume growth by all of the models and Sat-merged SIT. Moreover, all of these simulations reach their maximum SIV increase and decline in December and July, respectively. Compared to the model data without SIT assimilation (NAOSIM and PIOMAS), CMST and TOPAZ4 have more similar variability to Sat-merged SIT (shown in Fig. 1a and 1b). Nevertheless, it is impossible to identify the most accurate sea-ice simulation in this area due to the lag of *in-situ* observations.

## 5 Conclusions

In order to examine the sea ice volume variations in the Baffin Bay, we calculated the ensemble mean SIV fluxes together with their standard deviations and thermodynamic SIV growth from Sat-merged SIT and multi-model thickness data and NSIDC SID. Main conclusions can be summarized as follows:

(1) The sea ice volume (SIV) reaches its maximum in March or early April. It starts to increase from October until the onset of the melting season. The reduction occurs from May to September. The averaged maximum growth rate of $113\pm17$ km$^3$ month$^{-1}$ is found in December, while the maximum reduction rate of $-160\pm32$ km$^3$ month$^{-1}$ is in July.

(2) The annual mean SIV inflow and outflow are $411(\pm74)$ km$^3$ and $312(\pm80)$ km$^3$ year$^{-1}$, respectively. The SIV inflow in the melting season is only 9% of that in the freezing season. The SIV outflow in the melting season is a small fraction (11%) of the outflow in the freezing season.

(3) The maximum SIV freezing growth rate (67 km$^3$ month$^{-1}$) occurs in December while the maximum melting reduction rate ($-160$ km$^3$ month$^{-1}$) happens in July. On average, ice freezing (218 km$^3$) takes place from October to April while the ice melting ($-335$ km$^3$) occurs from May to September indicating that the surface seawater salinity may increase from October to April and decrease from May to September, correspondingly.

(4) The freshwater flux imported into the Labrador Sea derived from the sea-ice volume flux is about 250 km$^3$ year$^{-1}$ (i.e., 8 mSv) and large freshwater fluxes are found from February to April. The maximal freshwater flux is about 65 km$^3$ month$^{-1}$ (i.e., 25 mSv) and occurs in March.

*Data availability.* The CMST sea ice thickness and drift data can be download from https://doi.org/10.1594/PANGAEA.891475 (Mu et al., 2018b) and https://doi.org/10.1594/PANGAEA.906973 (Mu et al., 2019), respectively. The Polar Pathfinder Daily 25km EASE-Grid sea ice drift data are released by the National Snow and Ice Data Center (NSIDC, https://nsidc.org/data/nsidc-0116/versions/4, Tschudi et al., 2019; Tschudi et al., 2020). The PIOMAS sea ice thickness data are available at http://psc.apl.uw.edu/research/projects/arctic-sea-ice-

volume-anomaly/data/model_grid (Zhang and Rothrock, 2003). The TOPAZ4 sea ice data are available at http://marine.copernicus.eu (Xie et al., 2018).  The locally merged satellite sea ice data (Sat-merged SIT) can be obtained by connecting Jack C. Landy from University of Bristol.

*Author contributions.* QY conceptualized this study. CM carried out these estimations and wrote the paper. FF provided the NAOSIM sea ice data. All co-authors assisted during the writing process and critically discussed the contents.

*Competing interests.* The Authors declare that they have no conflict of interests.

*Acknowledgement.* This is a contribution to the Year of Polar Prediction (YOPP), a flagship activity of the Polar Prediction Project (PPP), initiated by the World Weather Research Programme (WWRP) of the World Meteorological Organization (WMO). We acknowledge the WMO WWRP for its role in coordinating this international research activity.  We thank the editor David Schroeder, referee Jack C. Landy and another anonymous referee for their constructive comments to improve the manuscript. Great gratitude is given to Jack C. Landy and Isolde Glissenaar from University of Bristol for processing the satellite-based sea ice data in the Baffin Bay and Jiping Xie from Nansen Environmental and Remote Sensing Center for providing the TOPAZ4 SIT data. Thanks are also given to Yongwu Xiu and Qian Shi from Sun Yat-sen University for their help of data processing.

*Financial support.* This study is supported by the National Natural Science Foundation of China (No. 41922044, 41941009, 41676185), the Guangdong Basic and Applied Basic Research Foundation (No. 2020B1515020025), the Fundamental Research Funds for the Central Universities (No. 19lgzd07).

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

45

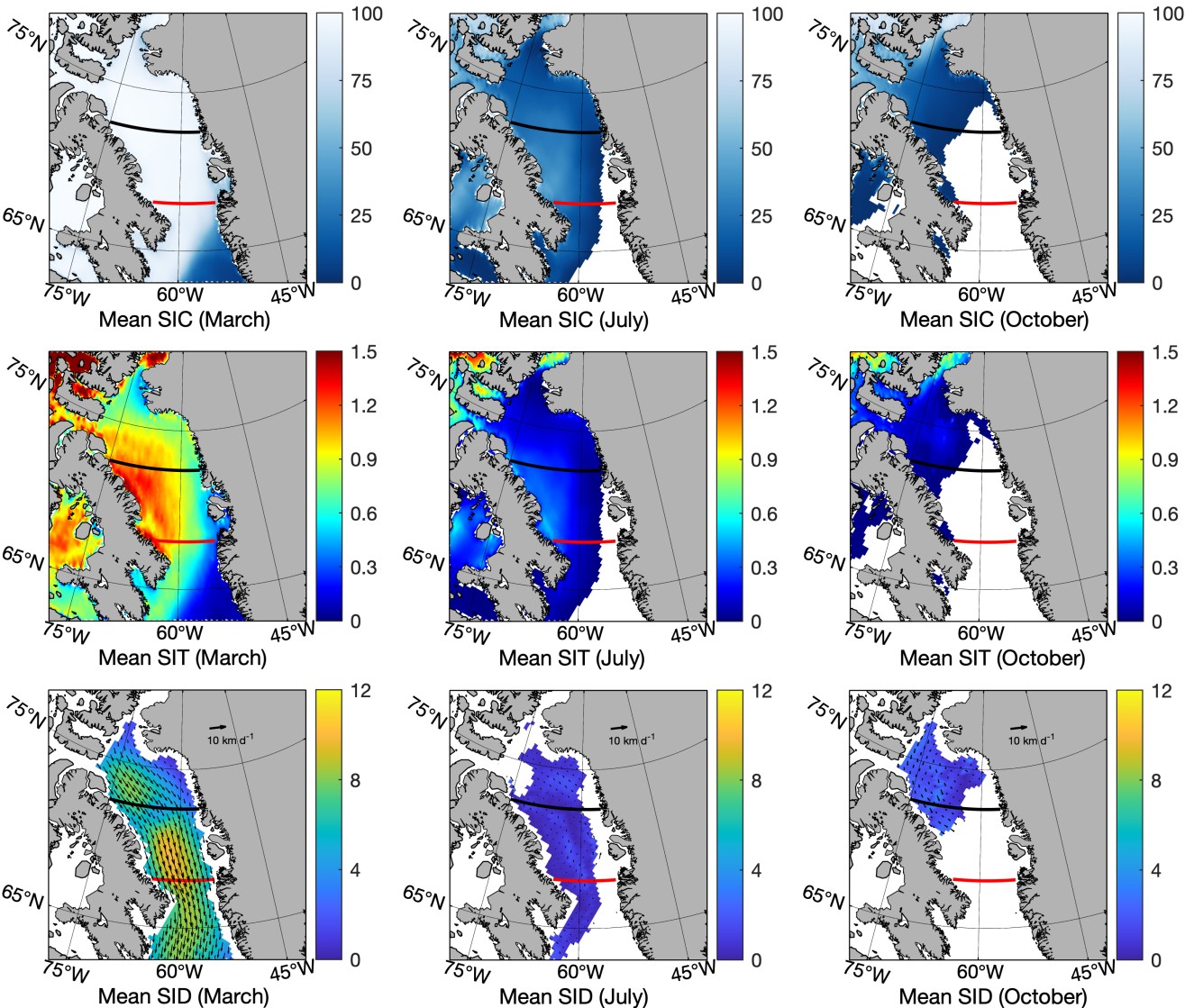

**Figure 1.** The ensemble mean sea ice concentration (top row: SIC, unit: %) and thickness (middle row: SIT, unit: m) in March, July, and October averaged from CMST, NAOSIM, PIOMAS and Sat-merged SIT over the period 2011-2016. Sea ice drift (bottom row: SID, unit: km d$^{-1}$) is calculated by averaging data from NSIDC. Note that the Sat-merged SIT in the ensemble are only valid in March and October.

The black line shows the SIV inflow gate, and the red line denotes the SIV outflow gate in the Baffin Bay.

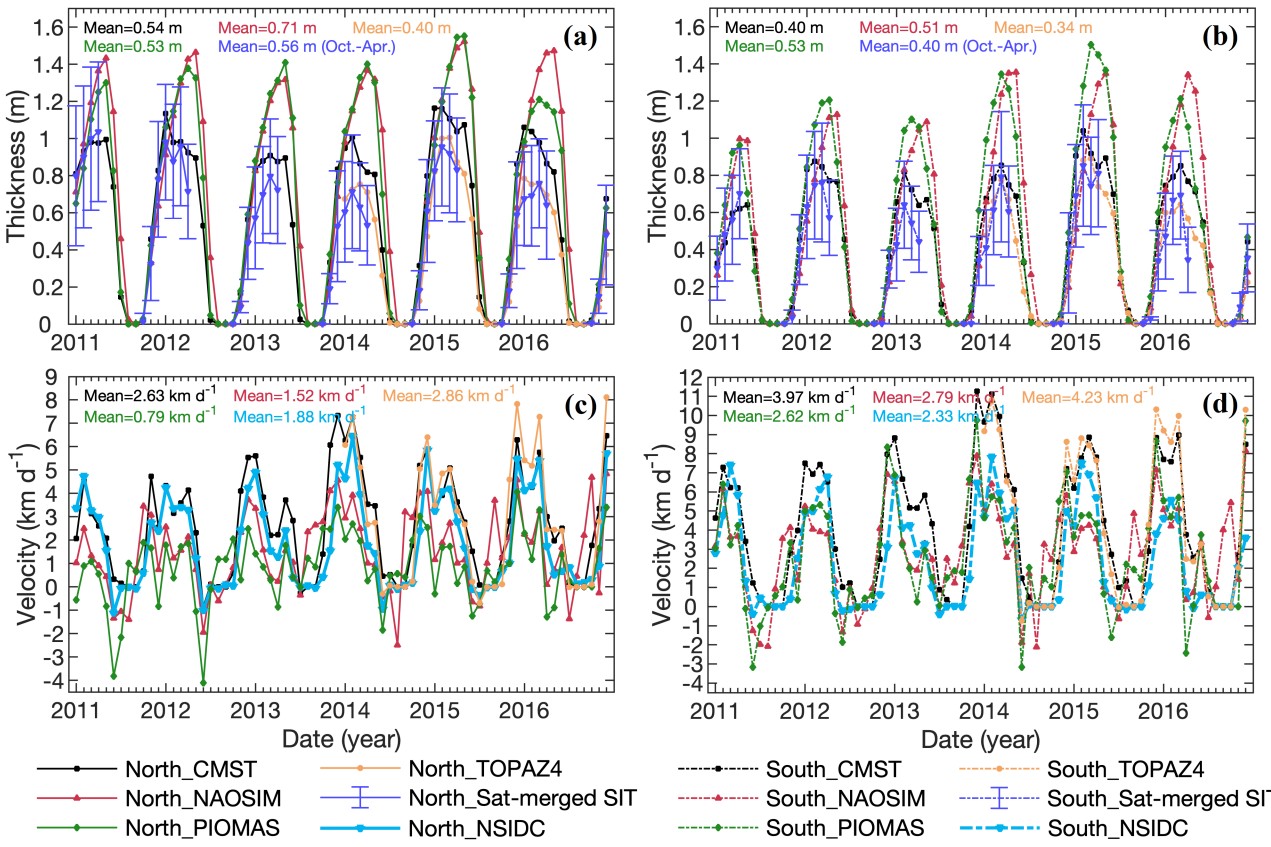

**Figure 2.** The monthly mean variations of sea ice thickness and southward velocity over the northern inflow gate and southern outflow gate (SIT: a and b, SID: c and d). The full lines in the left panel and dashed lines in the right panel represent sea ice variables over the north gate

and south gate, respectively. The different colours denote different input sea ice data. Note that the Sat-merged SIT with corresponding uncertainty is from a locally merged sea ice data in the Baffin Bay.

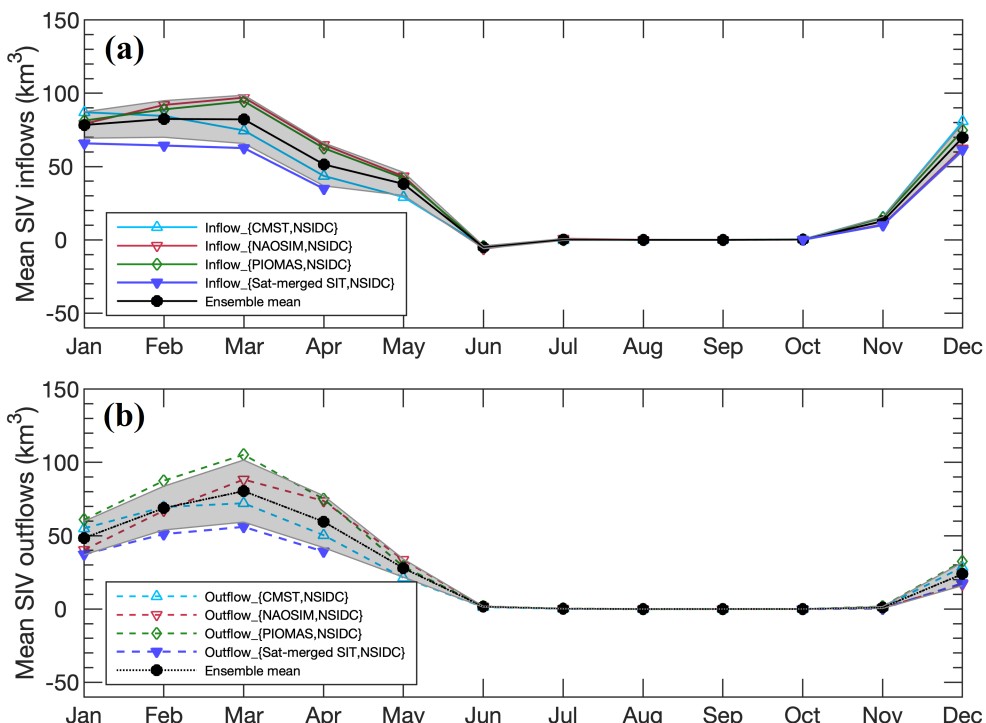

**Figure 3.** Averaged sea ice volume (SIV) (a) inflows through the north gate and (b) outflows through the south gate between 2011 and 2016. The cyan lines are the fluxes derived from CMST SIT and NSIDC SID, the red lines indicate estimates from NAOSIM SIT and NSIDC SID, the green lines denote the fluxes from PIOMAS SIT and NSIDC SID, the blue line is for the fluxes from Sat-merged SIT and NSIDC SID and the black lines represent the ensemble mean fluxes from the four inflows and outflows, respectively. Shaded areas indicate the standard deviation derived from the four different inflows and outflows, respectively.

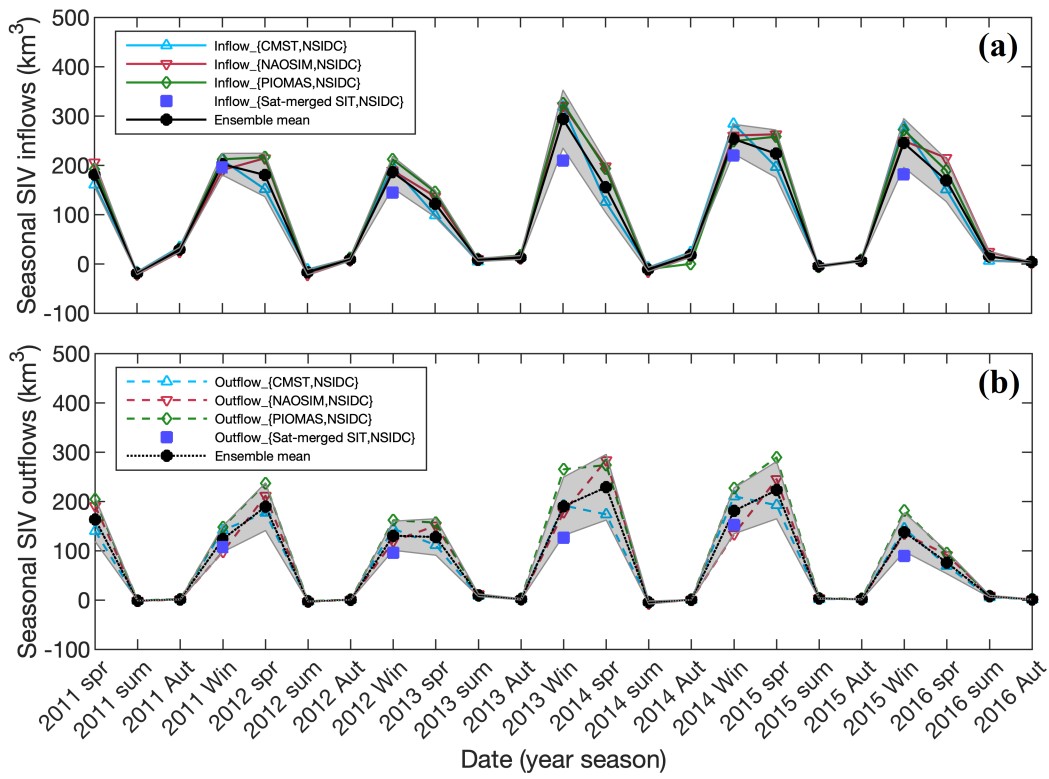

**Figure 4.** As Fig. 3 but for long-term seasonal evolution of sea ice inflows and outflows. Note that these blue squares represent the SIV fluxes from Sat-merged SIT and NSIDC SID.

70

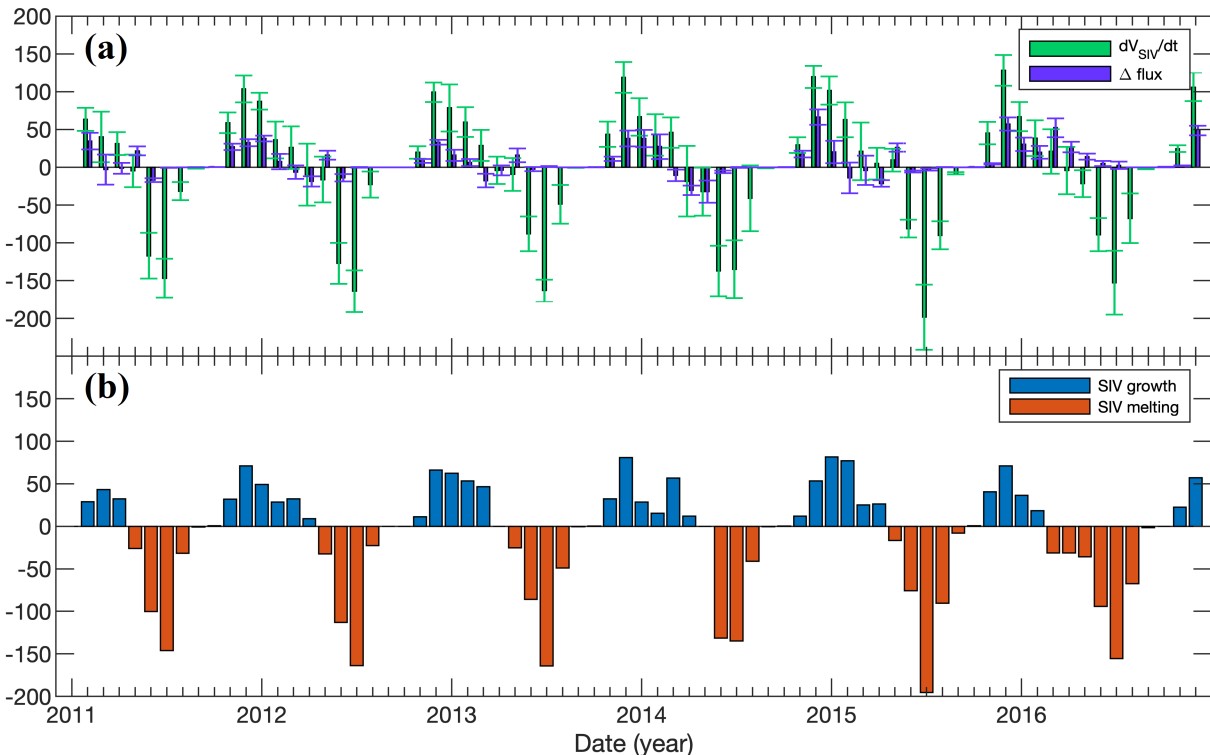

**Figure 5.** The ensemble mean sea ice volume changes from net ice flux and thermodynamics growth. (a) The ensemble mean SIV variability (dV$_{SIV}$/dt, green bar) in the defined Baffin Bay area and the net SIV flux (Δflux, purple bar) together with the ensemble spread (error bar). (b) The SIV variability derived from ice freezing (blue bar) and melting (orange bar) in the defined area.

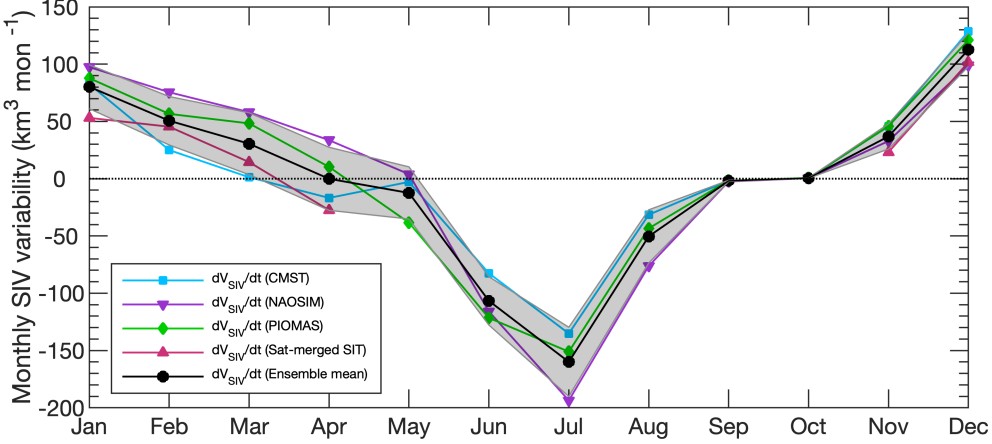

**Figure 6.** The sea ice volume changes from CMST (dV$_{SIV}$/dt (CMST), cyan line), NAOSIM (dV$_{SIV}$/dt (NAOSIM), purple line), PIOMAS (dV$_{SIV}$/dt (PIOMAS), green line), satellite observation (dV$_{SIV}$/dt (Sat-merged SIT), violet red line) and the ensemble mean (dV$_{SIV}$/dt (Ensemble mean), black line) in the Baffin Bay area. The shading indicates the ensemble spread (one standard deviation).

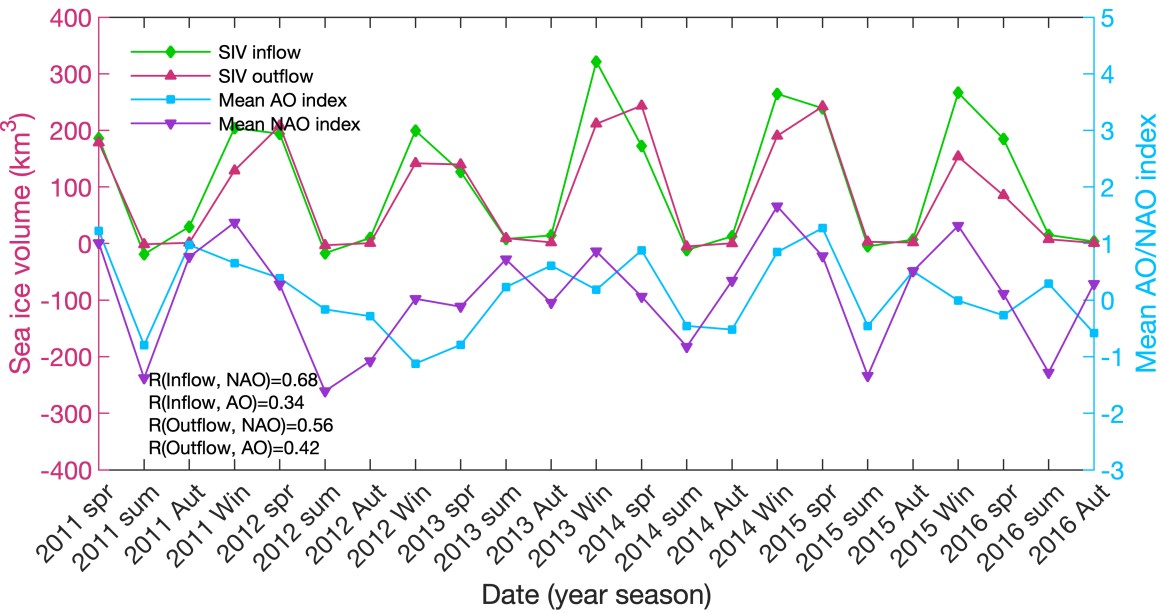

**Figure 7.** Time series of seasonal mean SIV inflow (green line), outflow (violet red line) in the Baffin Bay. The NAO (purple line) and AO (cyan line) indexes are averaged in the same period. R represents the correlation coefficient between NAO/AO and inflow and outflow.

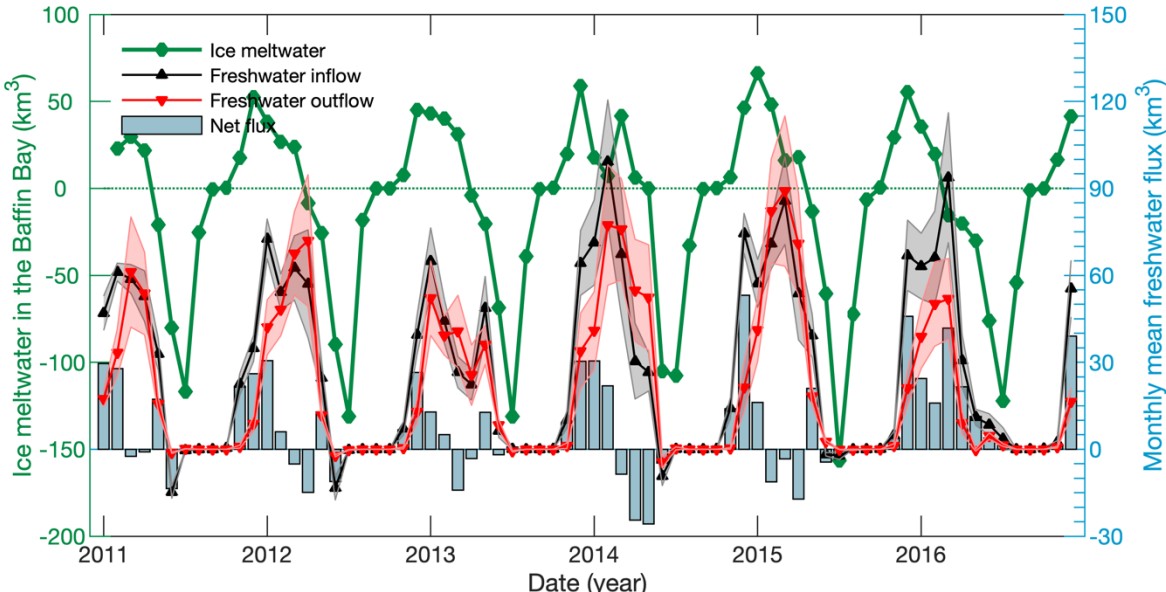

**Figure 8.** Freshwater from sea ice inflow (black line) through the north gate and outflow (red line) through the south gate (Davis Strait), and sea ice growth/melting (green line) in the Baffin Bay. The net flux of freshwater derived from net SIV flux (i.e., sea ice inflow minus outflow) are presented in skyblue bar.

**Table 1.** Monthly mean freshwater fluxes ($km^3$ $month^{-1}$) imported into the Labrador Sea that derive from the sea ice volume outflow.

|  | Jan | Feb | Mar | Apr | May | Jun | Jul | Aug | Sep | Oct | Nov | Dec |
|---|---|---|---|---|---|---|---|---|---|---|---|---|
| CMST_NSIDC | 44 | 56 | 58 | 40 | 17 | 1 | 0 | 0 | 0 | 0 | 1 | 23 |
| NAOSIM_NSIDC | 32 | 54 | 71 | 59 | 27 | 1 | 0 | 0 | 0 | 0 | 1 | 13 |
| PIOMAS_NSIDC | 49 | 70 | 84 | 60 | 23 | 1 | 0 | 0 | 0 | 0 | 1 | 26 |
| Sat-merged SIT_NSIDC | 30 | 41 | 45 | 31 | - | - | - | - | - | 0 | 0 | 14 |
| Ensemble mean | 39 | 55 | 65 | 48 | 22 | 1 | 0 | 0 | 0 | 0 | 1 | 19 |