# Peer review of "Ensemble-based estimation of sea-ice volume variations in the Baffin Bay"

_The Cryosphere, 2020_

## Referee Comment (RC1) · Jack Landy (Referee) · 5 Aug 2020

This study provides a more thorough assessment of annual sea ice volume changes in and solid ice freshwater flux variations across Baffin Bay than previous work. Combining several state-of-the-art sea ice models, some including data assimilation, enables the authors to estimate an uncertainty envelope around sea volume changes in the absence of in-situ or satellite observations. The amount of sea ice forming thermodynamically in Baffin Bay and the volume of freshwater exported from the bay into the Labrador Sea have critical downstream impacts on deepwater formation and the overturning circulation of the North Atlantic. So, I expect these results will be valued in the climate and physical oceanography communities.

[Figure]

I have made a few comments regarding the choice of datasets and methods used, particularly relating to use of only a single ice motion dataset and rejecting the use of satellite thickness observations. It would also be great to include more context for the calculated solid ice freshwater fluxes. Otherwise the manuscript seems to be in a good state and my remaining comments/edits are all quite minor. Please do get in contact if you have questions regarding these comments. Kind regards, Jack Landy

General comments:

1. Three model ice volume products are used but only one drift product. Alternative drift vectors from OSISAF and/or Kimura et al could also be used to improve the determination of the volume flux uncertainty envelope. Line 62, is OSISAF not available year-round in BB? If other products are not available year-round or have full coverage over BB, can you estimate the uncertainty envelope for the ice motion for the seasons/region where they do overlap and use that in your determination of overall error?

2. L59-60, in my opinion the SIT data from remotely-sensed observations have sufficient validity to compare with the model simulations. If there are clear biases that have been identified in Baffin Bay or in regions with similar sea ice regimes, then please discuss here. Otherwise I suggest to add a short comparison of the winter SIT evolution between the models, SMOS and CS2 or CS2SMOS, with the uncertainties of the observations illustrated, to gauge the validity of the models individually and as a collective. You may be able to discard one model in your ensemble, for instance, if it shows clear deviation from the satellite observations.

3. I recommend adding greater depth to the discussion on Baffin Bay/Labrador Sea freshwater budget. How do your results for the freshwater volume stored in ice within Baffin Bay compare to past estimates? How about the solid ice flux across Davis Strait? More importantly what is the context of the solid ice fluxes within the full freshwater budget? What

4. I would suggest having another careful check through the text, as there are quite a

few minor spelling mistakes and grammatical errors.

Minor comments/edits: Line 23. '. . .largest SIV outflow in spring of 2014' why?

L20. What about the freshwater budget? How much ice meltwater enters the ocean over the melt season? This is the key missing feature of the abstract, with respect to freshwater and deepwater formation.

L23. Draining off what? The Greenland Ice Sheet, liquid freshwater in the ocean proper, both..?

L31. Large errors with respect to what? Other regions or to other model-based thickness estimates?

L34-35. Can you define the directions of these fluxes?

L45-46. This argument requires more detailed explanation.

L50. I am not convinced the satellite based products are inappropriate to be used in this region. Can you provide an argument with supporting evidence why satellite measurements, including SMOS and/or altimetry, cannot be used here? (I do understand the satellite products only capture the winter ice growth season, so cannot be used to determine the full annual ice volume budget, which is in my mind a better reason not to use them than their apparently limiting uncertainties). You also state that the spatial distributions of model SIT are similar to that derived from satellites in Landy et al 2017; so why then are the satellite observations inappropriate to be used?

L53. Spell out the model acronyms.

L54-59. Please list the exact SIC, SIT and SST products used for assimilation into the models, as this clearly affects their interpretation.

L108. Why are the CryoSat-2 or CS2SMOS SIT data inappropriate in Baffin Bay? What does the strong seasonality have to do with it, and what do you mean by that?

L118. How were the drift observations validated? With in situ measurements?

L153. I do think it is worth including the CS2 or CS2SMOS cycle in your comparisons here.

Fig 2. Can you explain why the CMST simulations how a 'flattening off' of sea ice volume increase at the end of winter, when NAOSIM and PIOMAS are still rising?

L165. 'cycle' rather than 'trend'?

L193. What are the +/- as percentages?

L200. What do you mean by 'reach a maximum in spring/winter with a mean value of...'? Confusing

L205. Can you explain why a constant factor of 0.8 is used and justify it? (It is not sufficient just to include a citation without deeper explanation)

L206. Can you place this value of 271 km3 yr-1 in context? What is that in Sv? How does it compare with literature values for the net liquid FW flux across approx. the same southern gate between Baffin Bay and the Labrador Sea from other studies?

L235. It is unclear what you mean by 'We thus speculate that the thick ice is exported from the Arctic since the higher ice velocity is also found in these areas'. What point are you making?

L228-234. How do your results compare with the cited studies? Are the net volume growth/melting terms similar or very different (accounting for disparities in the study area)?

L242. How do you know the drift is underestimated? Have you tried comparing with another product, e.g. OSISAF for at least the months and time period they overlap?

---

## Referee Comment (RC2) · Anonymous Referee #2 · 25 Aug 2020

The sea ice volume variations within the Baffin Bay is investigated using mode-based sea ice thickness and NSIDC sea ice drift product. Since field measurements of sea ice thickness is scarce, this study presents the best way to estimate the sea ice in-flow/outflow of the bay. Moreover, the volume amounts in associated with freezing and melting processes are also quantified. Generally, this is a good attempt to conduct the studies related to sea ice volume, which is a better indicator, in relative to area, to interpret the current rapid climate changes. Before the publication of this study, minor revisions are needed as follows: L77, "...in Fram Strait and obtained" to "..in Fram Strait to obtain.' L79 "2.2..." L88 "2.3" L101 "..the years..." to " the short period" L110 " ...full filled..." to "... fully filled' L114 "date" to "data" L 140 " are a typical representation" to " typically represent" L 166 the sentence for " A fairly....ice drift". Make it to

two short sentence in order to clarify your description. L181 "Originate" to "Originates" L 186- 189, this sentence is too long to follow the content. I recommend the authors to make it short or by dividing it to two sentences. L197, remove ",respectively" L199 "In average" to "On average" L210 "invested" to "investigated"? L214, "in average" to "on average" L224 "found" to "identified" L226 what "these areas" represents for? Please clarify. L240 "with of the usage" to "with the usage"

Does the author have considered the impacts of large-scale atmospheric circulation, such as NAO, on the variations of sea ice volume in Baffin Bay? The NAO may be associated with the inflow/outflow, as well as the freezing and melting processes. Therefore, through the analysis of the correlation between NAO and sea ice volume changes owing to these processes may give us a preliminary understanding of the role of the large-scale atmospheric circulation in modulating the Baffin Bay sea ice volume variations.

---

## Author Comment (AC1) · 30 Oct 2020

**General Comments:**

This study provides a more thorough assessment of annual sea ice volume changes in and solid ice freshwater flux variations across Baffin Bay than previous work. Combining several state-of-the-art sea ice models, some including data assimilation, enables the authors to estimate an uncertainty envelope around sea volume changes in the absence of in-situ or satellite observations. The amount of sea ice forming thermodynamically in Baffin Bay and the volume of freshwater exported from the bay into the Labrador Sea have critical downstream impacts on deep water formation and the overturning circulation of the North Atlantic. So, I expect these results will be valued in the climate and physical oceanography communities.

I have made a few comments regarding the choice of datasets and methods used, particularly relating to use of only a single ice motion dataset and rejecting the use of satellite thickness observations. It would also be great to include more context for the calculated solid ice freshwater fluxes. Otherwise the manuscript seems to be in a good state and my remaining comments/edits are all quite minor.

Dear Reviewer:

  We would like to thank you for the constructive comments to improve this manuscript. We agree that adding more ice motion data and satellite thickness observations will improve the estimations of sea ice fluxes and the local ice volume variations. Thus, we added OSISAF drift and satellite-based sea ice thickness observations (Landy et al., 2017) to improve our estimations as suggested. Because the sea ice volume variations are also estimated based on the satellite observations in the revised text, the title of our manuscript has been be modified as "Ensemble-based estimation of sea-ice volume variations in the Baffin Bay". Furthermore, we discussed the freshwater budget in the Baffin Bay and Labrador Sea in the discussion part (Section 4) of our revised manuscript and we compared our estimate of the freshwater volume stored in ice with estimate of previous studies. Additionally, we included maps of the monthly mean freshwater fluxes as well as the freshwater volume that is stored in the sea ice in the bay in our revised manuscript.

  The specific responses and revisions are elucidated below shown in blue font for clarity.

Corresponding Author: Qinghua Yang
Email: yangqh25@mail.sysu.edu.cn

**Specific comments:**

**Point 1:** Three model ice volume products are used but only one drift product. Alternative drift vectors from OSISAF and/or Kimura et al could also be used to improve the determination of the volume flux uncertainty envelope. Line 62, is OSISAF not available year-round in BB? If other products are not available year-round or have full coverage over BB, can you estimate the uncertainty envelope for the ice motion for the seasons/region where they do overlap and use that in your determination of overall error?

*Response 1:* We agree with this constructive comment and added the well-validated low resolution OSISAF drift (OSI-405) to consolidate our estimates of sea-ice volume (SIV) fluxes in the bay. We choose the OSISAF drift rather than the KIMURA drift because the OSISAF and NSIDC drift data are proved to have valid performance in the Arctic (Sumata et al., 2014). OSISAF drift data performs better than other data (i.e., NSIDC, CERSAT and KIMURA), however, for the study period, OSISAF drift data is only available for freezing season (October-April) while the summer season drift data is provided since 2017 (http://www.osi-saf.org/?q=content/sea-ice-products). Moreover, because of the reduced spatial coverage of OSISAF drift data (OSI-405) in lower latitudes (e.g., along the south gate in this study), it may cause some unquantifiable errors in Baffin Bay if solely used. The coverage of OSISAF and NSIDC drift in the freezing season of 2011 is shown in Figure 1 as an example. Therefore, we averaged the OSISAF drift and the NSIDC drift during the freezing season to get an ensemble mean sea ice drift. For the melting season (May-September), we still only use the NSIDC drift.

All estimates of sea-ice volume (SIV) and of the fluxes have been recalculated with the satellite-based sea ice drift and thickness (will be discussed in Point 2) and the model ice thicknesses. The updated results (Fig. S1-S8 and Table S1) are shown in the supplement of this letter of response and these estimations are also updated in our revised manuscript. *(please see these updated results in our revised manuscript)*

[Figure]

**Figure 1.** Monthly mean sea ice drift in the freezing months of 2011 from (a) NSIDC and from (b) OSISAF

**Point 2:** L59-60, in my opinion the SIT data from remotely-sensed observations have sufficient validity to compare with the model simulations. If there are clear biases that have been identified in Baffin Bay or in regions with similar sea ice regimes, then please discuss here. Otherwise I suggest to add a short comparison of the winter SIT evolution between the models, SMOS and CS2 or CS2SMOS, with the uncertainties of the observations illustrated, to gauge the validity of the models individually and as a collective. You may be able to discard one model in your ensemble, for instance, if it shows clear deviation from the satellite observations.

*Respond 2:* We agree that the remotely-sensed observations have sufficient validity in the Arctic Basin. However, in Baffin Bay, SMOS SIT is proved to underestimate because (1) SMOS only provides the valid SIT that is thinner than 1 m and (2) the 100% ice concentration assumption in the retrieval algorithm is not well justified (Tian-Kunze et al., 2014; Tietsche et al., 2018). Moreover, sea ice in the Baffin Bay is dominated by

seasonal thin ice and CS2 has large uncertainty in the area where SIT is thinner than 1 m (Figure 2, Ricker et al., 2017). Based on the above discussion, we decided not to use SMOS and CS2 or CS2SMOS to estimate the sea ice fluxes in Baffin Bay.

Instead, we choose a local satellite-based SIT product that calculates SIT from CS2 radar freeboard together with passive microwave (PMW)-derived snow depth and merges it with SMOS where the CS2 derived thickness is thinner than 1 m (Landy et al., 2017; Landy et al., 2019; Landy et al., 2020). To improve the estimation of the volume flux, the satellite-based SIT is jointly used with CMST, NAOSIM and PIOMAS SIT in an ensemble approach.

**Point 3:** I recommend adding greater depth to the discussion on Baffin Bay/Labrador Sea freshwater budget. How do your results for the freshwater volume stored in ice within Baffin Bay compare to past estimates? How about the solid ice flux across Davis Strait? More importantly what is the context of the solid ice fluxes within the full freshwater budget?

*Respond 3:* We agreed that more detailed discussions on Baffin Bay/Labrador Sea freshwater budget are very necessary. We now estimated the monthly mean freshwater volumes derived from SIV inflow, outflow and the net SIV flux (i.e., SIV inflow minus outflow). The estimations were shown in Figure S7, including the freshwater volume variation from sea ice growth/melting processes in the Baffin Bay. Also, the freshwater volume stored in ice within the Baffin Bay and the solid ice flux across Davis Strait are further compared with previous studies. Then we compared the freshwater melting from sea ice with the full freshwater budget as suggested.

Some discussions are shown as follows:

*We further converted the monthly mean sea ice inflow, outflow, net flux as well as the ice growth/melting into the freshwater volumes (Fig. S7). It should be noted that the meltwater (from ice melting in the bay) inputed into the Baffin Bay reached its maximum of 156 km³ month⁻¹ (i.e., 58 mSv) in July of 2015 while the maximal rate of sea ice production reached 66 km³ month⁻¹ in January of 2015. The maximum volume of freshwater that stored in sea ice in the Baffin Bay is about 240 km³ in March/April. However, it is estimated with a maximum of 445 km³ in April by Landy et al. (2017). The smaller estimated freshwater storage may mostly attribute to the smaller study area in our defined area. Both of the peak freshwater inflow and outflow happen in the period of January-March. The maximal net freshwater entering into the Baffin Bay through the north gate is about 49 km³ month⁻¹ (i.e., 18 mSv) in December of 2014 while the maximum of freshwater flux derived from ice outflow that flowing into Labrador Sea is about 100 km³ month⁻¹ (i.e., 37 mSv) in May of 2014.*

*The freshwater flux through the Davis Strait ranges from 196 km³ (i.e.,6 mSv) in 2011 to 368 (i.e., 12 mSv) km³ in 2014. Annually, the mean freshwater flux derived from SIV outflow is about 287 km³ year⁻¹ (i.e., 9 mSv) which is slightly smaller than the estimation (10 mSv; Curry et al., 2014) based on ULS SIT observations. The comparable estimation indicates that our ensemble-based SIV fluxes seem to be reasonable and provide a new attempt to estimate the long-term SIV variation in the Baffin Bay restricted to the scarcity of SIT observations. Moreover, despite the*

*freshwater budget in Labrador Sea is affected by the sea ice freezing/melting, river and glacial input and net precipitation, the freshwater entering into the Labrador Sea from melt ice is only about 10% of the net liquid freshwater flux (93 mSv, Curry et al., 2014) through the Davis Strait.*
*(More details will be shown in the Discussion part of our revised manuscript)*

**Point 4:** I would suggest having another careful check through the text, as there are quite a few minor spelling mistakes and grammatical errors.
*Respond 4:* Thanks for this comment. We have carefully checked through this manuscript and we believe that the readability of our manuscript has been largely improved.

**Point 5:** Line 18: '…largest SIV outflow in spring of 2014' why?
*Respond 5:* We also noticed this difference (non-corresponding peaks) between inflows and outflows. This discordance can be attributed to the different time series of sea ice thickness (SIT) and drift along the north gate and the south gate. For example, the maximum of sea ice drift along the south gate reached its peak value during the winter of 2013, though the mean SIT during the winter of 2013 was relatively thinner than that in the spring of 2014.

**Point 6:** L20: What about the freshwater budget? How much ice meltwater enters the ocean over the melt season? This is the key missing feature of the abstract, with respect to freshwater and deep water formation.
*Respond 6:* As suggested, we added this key feature in the abstract:
*In the melt season, there is about 268 km$^3$ freshwater that melt from local sea ice entering the Baffin Bay. Annually, the mean fresh water that enters the Labrador Sea is about 287 km$^3$ year$^{-1}$ (i.e., 9 mSv), while it is only about 10% of the net liquid freshwater flux through the Davis Strait. The maximum fresh water flux peaks in March with the amount of 72 km$^3$ month$^{-1}$ (i.e., 27 mSv).*

**Point 7:** L23: Draining off what? The Greenland Ice Sheet, liquid freshwater in the ocean proper, both..?
*Respond 7:* Both the Greenland Ice Sheet glacial melt and liquid freshwater can affect the fresh water budget in the Baffin Bay. We realized the formerly ambiguous description. To avoid confusion, we modified our description as:
*This bay serves as an important pathway of southward flowing and cold freshwater draining off from the Arctic into the North Atlantic Oceans (Curry et al., 2010; Curry et al., 2014). Fresh water outflows through Davis Strait entering the Labrador Sea are integrated from Canadian Arctic Archipelago and west Greenland glacial runoff, river inputs, sea ice meltwater and precipitation (Curry et al., 2010; Curry et al., 2014; Tang et al., 2004).*

**Point 8:** L31: Large errors with respect to what? Other regions or to other model-based thickness estimates?

**Respond 8:** We refined this description as:

*Seasonal thin sea ice in the bay is dominating and satellite-based ice thickness has large errors in this bay with respect to other regions in the Arctic Basin.*

**Point 9:** L34-35. Can you define the directions of these fluxes?

**Respond 9:** We refined this sentence to *'In a recent study, Bi et al. (2019) analyzed the sea ice area fluxes in the Baffin Bay on a long-term time period for the first time and the increasing trend of the annual sea ice area flux are found, i.e., $38.9 \times 10^3$ km² decade$^{-1}$ for the inflows entering the bay through the north gate, $7.5 \times 10^3$ km² decade$^{-1}$ for the inflows through Lancaster Sound and $82.2 \times 10^3$ km² decade$^{-1}$ for the outflows through the south gate (Davis Strait), respectively.'*

**Point 10:** L45-46. This argument requires more detailed explanation.

**Respond 10:** We detailed this description as follows:

*The sea ice thermodynamic processes are closely related to the desalination of seawater and the freshwater budget in the Baffin Bay. For instance, during the sea ice freezing process the salt is discharged into the surface ocean water leading to a denser and saltier condition. Otherwise, when the sea ice starts melting during the melt season, the fresh/hyposaline water is drained into the surface water causing the desalination of surface sea water.*

**Point 11:** L50. I am not convinced the satellite based products are inappropriate to be used in this region. Can you provide an argument with supporting evidence why satellite measurements, including SMOS and/or altimetry, cannot be used here? (I do understand the satellite products only capture the winter ice growth season, so cannot be used to determine the full annual ice volume budget, which is in my mind a better reason not to use them than their apparently limiting uncertainties). You also state that the spatial distributions of model SIT are similar to that derived from satellites in Landy et al 2017; so why then are the satellite observations inappropriate to be used?

**Respond 11:** As we stated in response 2, we do agree with the referee that the remotely-sensed observations have sufficient validity in the Arctic Basin. However, in the Baffin Bay, SMOS SIT is proved to be underestimated because (1) SMOS only provides the valid SIT that thinner than 1 m and (2) the 100% ice concentration assumption during the data retrieval is not fully filed (Tian-Kunze et al., 2014; Tietsche et al., 2018). In addition, the sea ice in the Baffin Bay is dominated by seasonal thin ice, CS2 thus has large uncertainties in the area where SIT is thinner than 1 m (Figure 2, Ricker et al., 2017).

In a recent study of Landy et al. (2017), they developed a locally merged sea ice thickness data that calculated from CS2 radar freeboards and PMW snow depths, then merged with SMOS where the mean CS2 thickness is <1 m. This data is applied to estimate the sea ice variations in the Baffin Bay (Landy et al., 2017). Therefore, in the revised version of manuscript, we will add this data to calculate the sea ice volume fluxes and variations to improve the determination of the volume flux as suggested. The updated results are shown in the Supplement part of this response letter and they will

be also updated in our revised manuscript. *(please see these updated figures in the Supplement materials)*

**Point 12:** L53. Spell out the model acronyms.
*Respond 12:* We showed both the full name and acronyms of these reanalysis (i.e., combined model and satellite sea ice thickness (CMST), Pan-Arctic Ice Ocean Modeling and Assimilation System (PIOMAS), North Atlantic/Arctic Ocean Sea Ice Model (NAOSIM) and Towards an Operational Prediction system for the North Atlantic European coastal Zones (TOPAZ4)) in our revised version.

**Point 13:** L54-59. Please list the exact SIC, SIT and SST products used for assimilation into the models, as this clearly affects their interpretation.
*Respond 13:* Following your suggestions, we refined this sentence *"CMST is based on the Massachusetts Institute of Technology generation circulation model (MITgcm) and SMOS SIT from University of Hamburg, CryoSat-2 SIT from AWI and Special Sensor Microwave Imager/Sounder (SSMIS) ice concentration processed at IFREMER are assimilated (Mu et al., 2018a) while PIOMAS assimilates SIC from NSIDC near-real time product and sea surface temperature (SST) from the NCEP/NCAR Reanalysis (Zhang and Rothrock, 2003; Schweiger et al., 2011)."*.

**Point 14:** L108. Why are the CryoSat-2 or CS2SMOS SIT data inappropriate in Baffin Bay? What does the strong seasonality have to do with it, and what do you mean by that?
*Respond 14:* As we stated before, sea ice in the Baffin Bay is dominated by seasonal thin ice and CS2 has large uncertainty in the area where SIT is thinner than 1 m (Figure 2, Ricker et al., 2017). Moreover, in the Baffin Bay, SMOS SIT is proved to be underestimated because (1) SMOS only provides the valid SIT that thinner than 1 m and (2) the 100% ice concentration assumption during the data retrieval is not fully filed (Tian-Kunze et al., 2014; Tietsche et al., 2018). We realized the previous description is ambiguous, and we refined our description in the revised texts.

**Point 15:** L118. How were the drift observations validated? With in situ measurements?
*Respond 15:* The daily mean NSIDC drift observations are assessed with high-resolution (~ 100 m) Envisat wide-swath (~ 450 km) SAR observations and IABP buoy measurements (in the Arctic Ocean) from January 1979 to December 1994 (Bi et al., 2019). Their result shows (Figure 2, 3 and 4; Bi et al., 2019) that the NSIDC drift slightly underestimates the ice drift with a mean bias of -0.68 km day$^{-1}$ while has a high correlation (R=0.87) with SAR drift observation. To clarify our description, we refined this sentence as:
*Moreover, the NSIDC data set has been recently validated with high-resolution Envisat wide-swath SAR observations and IABP buoy measurements by Bi et al. (2019). Comparing with the SAR drift observation, the NSIDC drift slightly underestimates the ice drift with a mean bias of -0.68 km day$^{-1}$, while it has a high correlation (R=0.87)*

*with SAR drift (Bi et al., 2019).*

**Point 16:** L153. I do think it is worth including the CS2 or CS2SMOS cycle in your comparisons here.
*Response 16:* We agreed that CS2 and CS2SMOS have sufficient validity in the Arctic Basin. However, sea ice in the Baffin Bay is dominated by seasonal thin ice and CS2 has large uncertainty in the area where SIT is thinner than 1 m (Figure 2, Ricker et al., 2017). Instead of using CS2 or CS2SMOS, we decided to use a locally merged data that calculated from CS2 radar freeboards and PMW snow depths, then merged with SMOS where the mean CS2 thickness is <1 m (Landy et al., 2017; Landy et al., 2019; Landy et al., 2020). With this newly produced regional data, the disparities between satellite-observed SIT and modeled SITs are reduced. The comparisons are shown in Figure S2 (*also will be shown in Figure 2 in our revised manuscript*).

**Point 17:** Fig 2. Can you explain why the CMST simulations how a 'flattening off' of sea ice volume increase at the end of winter, when NAOSIM and PIOMAS are still rising?
*Response 17:* We also noticed these disparities between CMST, NAOSIM, PIOMAS and satellite-based observations, and this 'flattening off' also noticed by previous studies (Mu et al., 2018a; Tilling et al., 2015). For instance, the Northern hemisphere sea ice volumes from PIOMAS and CryoSat-2 reach their maximums in April and March, respectively (Figure 2, Tilling et al., 2015). The same phenomenon was also observed in TOPAZ4 system (personal communication with Jiping Xie on FAMOS meeting, 2018, Norway). The current clue we know is that both CMST and TOPAZ4 assimilate CS2/SMOS thickness observations, while NAOSIM and PIOMAS not. Further comprehensive diagnostics are needed to clarify it.

**Point 18:** L165. 'cycle' rather than 'trend'?
*Response 18:* Agreed, we used 'cycle' instead of 'trend' as suggested.

**Point 19:** L193. What are the +/- as percentages?
*Response 19:* The *'+/-number'* indicates one standard deviation among the ensemble members, i.e., inflows and outflows from (1) CMST SIT and observed SID, (2) NAOSIM SIT and observed SID, (3) PIOMAS SIT and observed SID and (4) observed SIT and observed SID. To specify this, we added the necessary description in section 2.7 as: *And we use one standard deviation (i.e., +/-number) among these ensemble members to show their dispersion degrees or the uncertainties of flux estimations in this study.*
Moreover, the reason that we did not applicate '+/-' as percentages is in order to keep consistent with previous studies (e.g., Bi et al., 2019; Min et al., 2019; Ricker et al., 2018; Spreen et al., 2020) so that the readers can easily compare these results.

**Point 20:** L200. What do you mean by 'reach a maximum in spring/winter with a mean value of…'? Confusing

*Response 20:* We rewrote our sentence with our updated results: *'On average, the maximum of ice inflows occurs in winter is with a mean value of 278(±46) km³ while ice outflows usually reach the maximums in winter/spring with a mean value of 186(±50)/184(±51) km³.'.*

**Point 21:** L205. Can you explain why a constant factor of 0.8 is used and justify it? (It is not sufficient just to include a citation without deeper explanation)

*Response 21:* We added some more explanation for the adoption of constant factor of 0.8 as:

*Furthermore, to quantify the freshwater imported into the Baffin Bay and Labrador Sea, where it is an important area of deep water formation, we convert the sea ice volume outflows to the fresh water fluxes by multiplying a factor (Spreen et al., 2020):*

$$(1 - \frac{S_{ice}}{S_{ref}}) \, (\frac{\rho_{ice}}{\rho_{water}}) \approx 0.8, \qquad\qquad (3)$$

*where sea ice salinity ($S_{ice}$) is assumed to be 4 psu, the reference seawater salinity $S_{ref}$ is 34.8 psu, sea ice density ($\rho_{ice}$) is 901.3 kg m-3 and seawater density ($\rho_{water}$) is 1023.9 kg m-3 (Haine et al., 2015; Serreze et al., 2006).*

**Point 22:** L206. Can you place this value of 271 km³ yr⁻¹ in context? What is that in Sv? How does it compare with literature values for the net liquid FW flux across approx. the same southern gate between Baffin Bay and the Labrador Sea from other studies?

*Response 22:* The results in our manuscript were updated following your suggestions. This sentence was also revised with our updated results: *Annually, the amount of freshwater flux that exported into the Labrador Sea derived from SIV flux is bout 287 km³ year ⁻¹ (i.e., 9 mSv). And the relatively large fresh water fluxes are found from January to April peaking at 72 km³ month⁻¹ (i.e., 27 mSv) in March. The annual mean freshwater derived from ice meltwater in previous studies is range from 10 mSv (i.e.,331 km³ year⁻¹; Curry et al., 2014) to 21.3 mSv (i.e., 873 km³ year⁻¹ of SIV; Tang et al., 2004) which is larger than our estimation.*
Further, the fresh water entering into the Labrador Sea that melts from sea ice is also discussed in our section 4 (Discussion section of our revised manuscript) as Point 3 described.

**Point 23:** L235. It is unclear what you mean by 'We thus speculate that the thick ice is exported from the Arctic since the higher ice velocity is also found in these areas'. What point are you making?

*Response 23:* We have refined our description as follows:
*And we notice that the ice thicker than 0.5 m is mostly located near the Nares Strait in October companying with higher ice velocity (more than 10 km day⁻¹) identified near the Smith Sound and Lancaster Sound by CMST (figure not shown). We thus speculate that most of the thick ice may be exported from the Arctic since the higher ice velocity is also found in the corresponding area of the thick ice located (i.e., Nares Strait), and*

*the faster ice is usually deemed to be a proxy for higher ice flux. This is also noticed in previous studies (e.g., Kwok, 2007, 2005).*

**Point 24:** L228-234. How do your results compare with the cited studies? Are the net volume growth/melting terms similar or very different (accounting for disparities in the study area)?

*Response 24:* We added these comparisons with Landy et al. (2017) as suggested:

*The annual mean rate of ice production in our study is 52 km³ month⁻¹ while it is about 87 km³ month⁻¹ estimated in previous study (Table 3, Landy et al., 2017). Also, the monthly mean sea ice volume variability in our study is smaller than that of Landy et al. (2017) which can be attributed to different area of the study regions.*

We needed to interpret that the net volume growth/melting terms in our study is very different from the previous study (Landy et al., 2017), because we excluded the net SIV flux ($Q_{net}$) in the calculation ($\frac{dV}{dt} = Q_{net} + (\frac{dV_{therm}}{dt} + \frac{dV_{resid}}{dt})$) while Landy et al. (2017) only calculated the regional SIV variation (assuming ice melts in situ). So, it is difficult to compare these two estimations in terms of net volume growth/melting. However, we further compared the satellite-based sea ice volume variation (provided by Landy et al.) with our modeled estimations (shown in Fig. S6). Then the ensemble-based SIV variation is averaged with the modeled results (i.e., CMST, NAOSIM and PIOMAS) and satellite observation. We also estimated the rate of ice production applying the similar method used in Landy et al. (Table 3, 2017), the rate of ice production is about 52 km³ month⁻¹ in our study while it is 87 km³ month⁻¹ (SR10 adding SR11) in Landy et al. (2017). Because the study area (SR10 adding SR11) in Landy et al. (2017) is much larger than ours, this disparity between these two studies can be mostly attributed to different defined area.

**Point 25:** L242. How do you know the drift is underestimated? Have you tried comparing with another product, e.g. OSISAF for at least the months and time period they overlap?

*Response 25:* It is proved that NSIDC drift present a mean bias of -0.68 km day⁻¹ compared with SAR ice drift indicating that NSIDC drift is slightly slower (Figure 4, Bi et al., 2019). Also, Sumata et al., (2014) found that the monthly mean NSIDC drift was slightly slower than OSISAF drift and the spatial mean ice drift speed in the Arctic Ocean was also slightly slower than OSISAF, IABP/D and KIMURA ice drift data. Moreover, the sea ice volume export through the Fram Strait based on AWI CS2 sea ice thickness and NSIDC drift (version 3) shows a mean difference about -26 % comparing with that based on AWI CS2 sea ice thickness and OSISAF drift. However, the new version of NSIDC SID performs much better according to Sumata (pers. Com.).

NSIDC drift is not compared with OSISAF drift in the previous version of this manuscript. So, following the referee's suggestions, we further added OSISAF drift into the intercomparison between NSIDC, CMST, PIOMAS, NAOSIM and TOPAZ4 ice drift. In our intercomparison (Figure S2), the averaged SID (averaged SID from OSISAF and NSIDC drift during the freezing season) is larger than the NSIDC drift

(compared with the mean NSIDC drift in our previous manuscript). And then we used the averaged SID to calculated the four ice flux estimations (i.e., (1) CMST SIT and Sat-SID, (2) NAOSIM SIT and Sat-SID, (3) PIOMAS SIT and Sat-SID and (4) Sat-merged SIT and Sat-SID) during the freezing season as the referee suggested.

**Reference**

Bi, H., Zhang, Z., Wang, Y., Xu, X., Liang, Y., Huang, J., Liu, Y., and Fu, M.: Baffin Bay sea ice inflow and outflow: 1978– 1979 to 2016–2017, The Cryosphere, 13, 1025-1042, doi: 10.5194/tc-13-1025-2019, 2019.

Landy, J., C., Petty, A. A., Tsamados, M., and Stroeve, J. C.: Sea ice roughness overlooked as a key source of uncertainty in CryoSat-2 ice freeboard retrievals. J. Geophys. Res-Oceans., 125, e2019JC015820. doi: 10.1029/2019JC015820, 2020.

Landy, J., C., Tsamados, M., and Scharien, R., K.: A Facet-Based Numerical Model for Simulating SAR Altimeter Echoes From Heterogeneous Sea Ice Surfaces, IEEE T. Geosci. Remote , 57, 4164-4180, doi: 10.1109/TGRS.2018.2889763, 2019.

Landy, J., C., Ehn, J. K., Babb, D. G., Thériault, N., and Barber, D. G.: Sea ice thickness in the Eastern Canadian Arctic: Hudson Bay Complex & Baffin Bay, Remote Sens. Environ., 200, 281-294, doi: 10.1016/j.rse.2017.08.019, 2017.

Curry, B., Lee, C. M., and Petrie, B.: Volume, Freshwater, and Heat Fluxes through Davis Strait, 2004–05, J. Phys. Oceanogr., 41, 429-436, doi: 10.1175/2010JPO4536.1, 2010.

Curry, B., Lee, C. M., Petrie, B., Moritz, R. E., and Kwok, R.: Multiyear Volume, Liquid Freshwater, and Sea Ice Transports through Davis Strait, 2004–10, J. Phys. Oceanogr., 44, 1244-1266, doi: 10.1175/JPO-D-13-0177.1, 2014.

Day, J. J., Hawkins, E., and Tietsche, S.: Will Arctic sea ice thickness initialization improve seasonal forecast skill? Geophys. Res. Lett., 41, 7566–7575, doi: 10.1002/2014GL061694, 2014.

Haine, T. W. N., Curry, B., Gerdes, R., Hansen, E., Karcher, M., Lee, C., Rudels, B., Spreen, G., de Steur, L., Stewart, K. D., and Woodgate, R.: Arctic freshwater export: Status, mechanisms, and prospects, Global Planet. Change, 125, 13-35, doi: 10.1016/j.gloplacha.2014.11.013, 2015.

Johnson, M., Proshutinsky, A., Aksenov, Y., Lindsay, A. T. N. R., Haas, C., Zhang, J., Diansky, N., Kwok, R., Maslowski, W, Häkkinen, S.,  Ashik, I., Cuevas, B.: Evaluation of Arctic sea ice thickness simulated by Arctic Ocean Model Intercomparison Project models, J. Geophys. Res-Oceans., 117, C00D13, doi: 10.1029/2011JC007257, 2012.

Kwok, R.: Baffin Bay ice drift and export: 2002–2007, Geophys. Res. Lett., 34, L19501, doi:10.1029/2007GL031204, 2007.

Kwok, R.: Variability of Nares Strait ice flux, Geophys. Res. Lett., 32, L24502, doi:10.1029/2005GL024768, 2005.

Min, C., Mu, L., Yang, Q., Ricker, R., Shi, Q., Han, B., Wu, R., and Liu, J.: Sea ice export through the Fram Strait derived from a combined model and satellite data set, The Cryosphere, 13, 3209-3224, doi: 10.5194/tc-13-3209-2019, 2019.

Mu, L., Losch, M., Yang, Q., Ricker, R., Losa, S. N., and Nerger, L.: Arctic-Wide Sea Ice Thickness Estimates from Combining Satellite Remote Sensing Data and a Dynamic Ice-Ocean Model with Data Assimilation During the CryoSat-2 Period, J. Geophys. Res-Oceans., 123, 7763-7780, doi: 10.1029/2018JC014316, 2018a.

Mu, L., Yang, Q., Losch, M., Losa, S. N., Ricker, R., Nerger, L., and Liang, X.: Improving sea ice thickness estimates by assimilating CryoSat-2 and SMOS sea ice thickness data simultaneously, Q. J. Roy. Meteor. Soc., 144, 529–538, https://doi.org/10.1002/qj.3225, 2018b.

Ricker, R., Girard-Ardhuin, F., Krumpen, T., and Lique, C.: Satellite-derived sea ice export and its impact on Arctic ice mass balance, The Cryosphere, 12, 3017-3032, doi: 10.5194/tc-12-3017-2018, 2018.

Ricker, R., Hendricks, S., Helm, V., Skourup, H., and Davidson, M.: Sensitivity of CryoSat-2 Arctic sea-ice freeboard and thickness on radar-waveform interpretation, The Cryosphere, 8, 1607-1622, doi: doi:10.5194/tc-8-1607-2014, 2014.

Ricker, R., Hendricks, S., Kaleschke, L., Tian-Kunze, X., King, J., and Haas, C.: A weekly Arctic sea-ice thickness data record from merged CryoSat-2 and SMOS satellite data, The Cryosphere, 11, 1607-1623, doi: 10.5194/tc-11-1607-2017, 2017.

Serreze, M. C., Barrett, A. P., Slater, A. G., Woodgate, R. A., Aagaard, K., Lammers, R. B., Steele, M., Moritz, R., Meredith, M., and Lee, C. M.: The large-scale freshwater cycle of the Arctic, J. Geophys. Res-Oceans., 111, C11010, doi: 10.1029/2005JC003424, 2006.

Schweiger, A., Lindsay, R., Zhang, J., Steele, M., Stern, H., and Kwok, R.: Uncertainty in modeled Arctic sea ice volume, J. Geophys. Res-Oceans., 116, C00D06, doi:10.1029/2011JC007084, 2011.

Spreen, G., de Steur, L., Divine, D., Gerland, S., Hansen, E., & Kwok, R.: Arctic sea ice volume export through Fram Strait from 1992 to 2014, J. Geophys. Res-Oceans., 125, e2019JC016039, doi: 10.1029/2019JC016039, 2020.

Tang, C. C. L., Ross, C. K., Yao, T., Petrie, B., DeTracey, B. M., and Dunlap, E.: The circulation, water masses and sea-ice of Baffin Bay, Prog. Oceanogr., 63, 183-228, doi: 10.1016/j.pocean.2004.09.005, 2004.

Tilling, R. L., Ridout, A., Shepherd, A., and Wingham, D. J.: Increased Arctic sea ice volume after anoma- lously low melting in 2013, Nat. Geosci., 8, 643–646, https://doi.org/10.1038/Ngeo2489, 2015.

Xiu, Y., Min C., Xie, J., Mu, L., Wu, X., Han, B., Yang, Q.: Evaluation of the Sea Ice Thickness in an Arctic Sea Ice-Ocean Reanalysis, J. Glaciol., in review, 2020.

Yang, Q., Losa, S. N., Losch, M., Jung, T., Nerger, L.: The role of atmospheric uncertainty in Arctic summer sea ice data assimilation and prediction, Q. J. Roy. Meteor. Soc., 141(691), 2314–2323, doi: 10.1002/qj.2523, 2015a.

Yang, Q., Losa, S. N., Losch, M., Liu, J., Zhang, Z., Nerger, L., Yang, H.: Assimilating summer sea-ice concentration into a coupled ice–ocean model using a LSEIK filter, Ann. Glaciol., 56(69), 38–44, doi: 10.3189/2015AoG69A740, 2015b.

Yang, Q., Losch, M., Losa, S. N., Jung, T., Nerger, L., Lavergne, T.: Brief

communication: The challenge and benefit of using sea ice concentration satellite data products with uncertainty estimates in summer sea ice data assimilation, The Cryosphere, 10(2), 761–774, doi: 10.5194/tc-10-761-2016, 2016.

Warren, S. G., Rigor, I. G., Untersteiner, N., Radionov, V. F., Bryazgin, N. N., Aleksandrov, Y. I., and Colony, R.: Snow depth on Arctic sea ice, J. Climate, 12, 1814–1829, 1999.

Zhang, J. and Rothrock, D. A.: Modeling Global Sea Ice with a Thickness and Enthalpy Distribution Model in Generalized Curvilinear Coordinates, Mon. Wea. Rev., 131, 845-861, doi: 10.1175/1520-0493(2003)131<0845:MGSIWA>2.0.CO;2, 2003.

**Supplement**

[Figure]

**Figure S1.** The ensemble mean sea ice concentration (top row: SIC, unit: %) and thickness (middle row: SIT, unit: m) in March, July, and October averaged over the period 2011-2016. Sea ice drift (bottom row: SID, unit: km d⁻¹) is calculated by averaging data from NSIDC and OSISAF. Note that the Sat-merged SIT and OSISAF drift in the ensemble are only valid in March and October. The black line shows the SIV inflow gate, and the red line denotes the SIV outflow gate in the Baffin Bay.

[Figure]

**Figure S2.** The monthly mean variations of sea ice thickness and southward velocity over the northern inflow gate and southern outflow gate (SIT: a and b, SID: c and d). The full lines in the left panel and dashed lines in the right panel represent sea ice inflow and outflow, respectively. The different colours denote different input sea ice data. Note that the Sat-merged SIT with corresponding uncertainty is from a regional merged sea ice data in the Baffin Bay and the Sat-SID is averaged from NSIDC and OSISAF drift.

[Figure]

**Figure S3.** Averaged sea ice volume (SIV) (a) inflows through the north gate and (b) outflows through the south gate between 2011 and 2016. The cyan lines are the fluxes derived from CMST SIT and Sat-SID, the red lines

indicate estimations from NAOSIM SIT and Sat-SID, the green lines denote the fluxes from PIOMAS SIT and Sat-SID, the blue line is for the fluxes from Sat-merged SIT and Sat-SID and the black lines represent the ensemble mean fluxes from the four inflows and outflows, respectively. Shaded areas indicate the standard deviation derived from the four different inflows and outflows, respectively.

[Figure]

**Figure S4.** As Fig. 3 but for long-term seasonal evolution of sea ice inflows and outflows. Note that these blue squares represent the SIV fluxes from Sat-merged SIT and Sat-SID.

[Figure]

**Figure S5.** The ensemble mean sea ice volume changes from net ice flux and thermodynamics growth. (a) The ensemble mean SIV variability (dVSIV/dt, green bar) in the defined Baffin Bay area and the net SIV flux (Δflux, purple bar) together with the ensemble spread (error bar). (b) The SIV variability derived from ice freezing (blue bar) and melting (orange bar) in the defined area.

[Figure]

**Figure S6.** The sea ice volume changes from CMST (dVSIV/dt (CMST), cyan line), NAOSIM (dVSIV/dt (NAOSIM), purple line), PIOMAS (dVSIV/dt (PIOMAS), green line), satellite observation (dVSIV/dt (Sat-merged SIT), violet red line) and the ensemble mean (dVSIV/dt (Ensemble mean), black line) in the Baffin Bay area. The shading indicates the ensemble spread (one standard deviation).

[Figure]

**Figure S7.** Freshwater from sea ice inflow (black line) through the north gate and outflow (red line) through the south gate (Davis Strait), and sea ice growth/melting (green line) in the Baffin Bay. The net flux of fresh water derived from net SIV flux (i.e., sea ice inflow minus outflow) are presented in skyblue bar.

[Figure]

**Figure S8.** Time series of seasonal mean SIV inflow (green line), outflow (violet red line) in the Baffin Bay. The NAO (purple line) and AO (cyan line) indexes are averaged in the same period. R represents the correlation coefficient between NAO/AO and inflow and outflow.

**Table S1.** Monthly mean fresh water fluxes ($km^3$ $month^{-1}$) imported into the Labrador Sea that derived from the sea ice volume inflows.

|  | Jan | Feb | Mar | Apr | May | Jun | Jul | Aug | Sep | Oct | Nov | Dec |
|---|---|---|---|---|---|---|---|---|---|---|---|---|
| CMST_Sat-SID | 55 | 63 | 65 | 45 | 17 | 1 | 0 | 0 | 0 | 0 | 1 | 31 |
| NAOSIM_ Sat-SID | 40 | 61 | 79 | 66 | 27 | 1 | 0 | 0 | 0 | 0 | 1 | 18 |
| PIOMAS_ Sat-SID | 61 | 80 | 94 | 66 | 23 | 1 | 0 | 0 | 0 | 0 | 2 | 35 |
| Sat-merged SIT _ Sat-SID | 37 | 46 | 50 | 34 | - | - | - | - | - | 0 | 0 | 19 |
| Ensemble mean | 48 | 62 | 72 | 53 | 23 | 1 | 0 | 0 | 0 | 0 | 1 | 26 |

---

## Author Comment (AC2) · 30 Oct 2020

**General Comments:**

The sea ice volume variations within the Baffin Bay is investigated using model-based sea ice thickness and NSIDC sea ice drift product. Since field measurements of sea ice thickness is scarce, this study presents the best way to estimate the sea ice inflow/outflow of the bay. Moreover, the volume amounts in associated with freezing and melting processes are also quantified. Generally, this is a good attempt to conduct the studies related to sea ice volume, which is a better indicator, in relative to area, to interpret the current rapid climate changes.

Dear Reviewer:

We would like to thank you for the helpful comments to improve this manuscript. Following your suggestions, we calculated the correlations between NAO/AO and sea-ice volume and sea-ice volume fluxes in Baffin Bay. As suggested by another referee, we further added the locally merged SIT observations and OSISAF ice drift to improve the estimation of the volume fluxes. We also revised the spelling mistakes and grammatical errors.

Below, we repeat each comment and insert our replies in the text where revisions were made. All responses are in blue font for clarity.

Corresponding Author: Qinghua Yang
Email: yangqh25@mail.sysu.edu.cn

**Specific comments:**

**Point 1:** L77, ". . .in Fram Strait and obtained" to "..in Fram Strait to obtain..'

*Response:* We realized that it was ambiguous in the previous version. We changed this sentence to *'Additionally, CMST is successfully applied to obtain a relatively accurate estimation of the year-round sea ice volume export through the Fram Strait (Min et al., 2019).'*

**Point 2:** L79 "2.2…" L88 "2.3"

*Response:* Thanks for your conscientious review of this manuscript. We revised this in our revised manuscript.

**Point 3:** L101 ".. the years…" to " the short period".

*Response:* Following your advice, we changed this sentence to: *Since the TOPAZ4 reanalysis data cover a short period from 2014 to 2018, the TOPAZ4 SIT and SID are only used for inter-comparison with the other sea ice data but not for any volume or*

*flux calculations in this study.*

**Point 4:** L110 " … full filled…" to "… fully filled'
*Response:* We have changed the "… full filled" to "… fully filled".

**Point 5:** L114 "date" to "data"
*Response:* We changed the "date" to "data".

**Point 6:** L140 " are a typical representation" to " typically represent"
*Response:* Thanks for your comments. We modified this sentence: *'We have chosen these months as they typically represent the seasonal cycle'.*

**Point 7:** L166 the sentence for " A fairly … ice drift". Make it to two short sentence in order to clarify your description.
*Response:* Thanks. Following your advice, we split this sentence into two short sentence: *'A fairly similar cycle of SID is shown by CMST, TOPAZ4 and satellite-based observation. However, both CMST and TOPAZ4 present a higher ice velocity than that from satellite-based observation while NAOSIM and PIOMAS underestimate the monthly mean ice drift'.*

**Point 8:** L181 "Originate" to "Originates".
*Response:* We modified this as suggested.

**Point 9:** L186-189, this sentence is too long to follow the content. I recommend the authors to make it short or by dividing it to two sentences.
*Response:* Thank you for this suggestion. We modified our description as follows:
*Landy et al. (2017) developed a 14-year SIT data in the eastern Canadian Arctic (ECA) from ICESat, CryoSat-2 and passive microwave (PMW) snow depths, then merged with SMOS where the mean CryoSat-2 thickness is <1 m. This satellite-based data is successfully utilized to calculate the local sea ice volume variation in the Baffin Bay while the sea ice volume fluxes and thermodynamic growth are not involved (Landy et al., 2017). Here we further calculated the SIV inflows, outflows and thermodynamic SIV growth from 2011 to 2016 with the usage of sufficiently validated CMST, widely used PIOMAS, parameter-optimized NAOSIM and satellite-based SIT data.*

**Point 10:** L197, remove ", respectively".
*Response:* We removed ", respectively" as suggested.

**Point 11:** L199 "In average" to "On average"
*Response:* Thanks, we have changed the "In average" to "On average".

**Point 12:** L210 "invested" to "investigated"?
*Response:* We have modified this sentence: *In this study, the locally thermodynamic processes are further investigated with the consideration of sea ice freezing, melting*

*and volume fluxes (Fig. 5).*

**Point 13:** L214, "in average" to "on average"
*Response:* We changed the "In average" to "On average".

**Point 14:** L224 "found" to "identified"
*Response:* We changed "found" to "identified" as suggested.

**Point 15:** L226 what "these areas" represents for? Please clarify.
*Response:* We changed our expression to: *We thus speculate that the thick ice is exported from the Arctic since the higher ice velocity is also found in the corresponding area of the thick ice located (i.e., Nares Strait), and the faster ice is usually deemed to be a proxy for higher ice flux.*

**Point 16:** L240 "with of the usage" to "with the usage".
*Response:* Thanks. We changed "with of the usage" to "with the usage".

**Point 17:** Does the author have considered the impacts of large-scale atmospheric circulation, such as NAO, on the variations of sea ice volume in Baffin Bay? The NAO may be associated with the inflow/outflow, as well as the freezing and melting processes. Therefore, through the analysis of the correlation between NAO and sea ice volume changes owing to these processes may give us a preliminary understanding of the role of the large-scale atmospheric circulation in modulating the Baffin Bay sea ice volume variations.
*Response:* Thank you for this constructive advice. We added an analysis on the correlation between NAO/AO and sea ice volume changes as suggested. The correlation coefficient (CC) between NAO/AO and SIV inflow and outflow for seasonal data are shown in Figure 1. The CCs between NAO and SIV inflow and outflow are 0.67 and 0.58, respectively. For AO and SIV inflow, the CCs are 0.31 and 0.37, respectively. However, we know that the long-term (climatic) time series of sea ice fluxes are required to substantiate these findings.
As suggested, we added this discussion in the discussion section.

[Figure]

Figure 1 Time series of seasonal mean sea ice volume (SIV) inflow (green line), outflow (violet red line) in the Baffin Bay. The NAO (purple line) and AO (cyan line) indexes are averaged in the same period. R represents the correlation coefficient between NAO/AO and inflow and outflow.

**Reference**

Landy, J. C., Ehn, J. K., Babb, D. G., Thériault, N., and Barber, D. G.: Sea ice thickness in the Eastern Canadian Arctic: Hudson Bay Complex & Baffin Bay, Remote Sens. Environ., 200, 281-294, doi: 10.1016/j.rse.2017.08.019, 2017.

Min, C., Mu, L., Yang, Q., Ricker, R., Shi, Q., Han, B., Wu, R., and Liu, J.: Sea ice export through the Fram Strait derived from a combined model and satellite data set, The Cryosphere, 13, 3209-3224, doi: 10.5194/tc-13-3209-2019, 2019.

---

## Author Response (AR1)

**Cover letter**

**Dear Editor:**

We thank you and two anonymous reviewers very much for the constructive comments and suggestions for the paper 'Ensemble-based estimation of sea-ice volume variations in the Baffin Bay' submitted to *the Cryosphere*. They are very valuable and very helpful for improving our manuscript. We have made a substantial revision according to the comments and suggestions from the editor and the two reviewers, and replied to them one by one below.

Qinghua Yang

On behalf of all the authors

**Responses to the editor**

Dear editor,

We appreciate your great efforts to improve this manuscript. Followed your comments and also the #1 comment from referee one, we have used OSISAF drift to calculate the sea ice volume (SIV) fluxes individually. However, we found that the mean southward and northward velocity from OSI-405 are much higher than NSIDC drift (V4) though the annual cycles of two drift data are similar (shown in Figure 1 in Response letter #1). This overestimation can be attributed to its (OSI-405) reduced spatial coverage in lower latitudes and rough mesh grid (62.5 km) in the Baffin bay (e.g., especially along the south gate in this study), because this causes only a few sea ice floes with higher velocity located in the center of the Davis Strait (i.e., south gate) can be measured. This would induce some unquantifiable errors calculating SIV fluxes in Baffin Bay if OSISAF drift is solely used. We also tried to average the NSIDC drift and OSISAF drift, but systematic differences between them are found. Moreover, the new version of NSIDC drift performs much better than the previous version (Hiroshi Sumata and Frank Kauker, personal communication). So, based on the above reasons, we decide to only use NSIDC drift to estimate the sea ice volume fluxes in the Baffin Bay.

Again, we thank you very much for your time and great efforts to improve this manuscript.

Qinghua Yang On behalf of all the authors

**Responses to referee #1**

**General Comments:**

This study provides a more thorough assessment of annual sea ice volume changes in and solid ice freshwater flux variations across Baffin Bay than previous work. Combining several state-of-the-art sea ice models, some including data assimilation, enables the authors to estimate an uncertainty envelope around sea volume changes in the absence of in-situ or satellite observations. The amount of sea ice forming thermodynamically in Baffin Bay and the volume of freshwater exported from the bay into the Labrador Sea have critical downstream impacts on deep water formation and the overturning circulation of the North Atlantic. So, I expect these results will be valued in the climate and physical oceanography communities.

I have made a few comments regarding the choice of datasets and methods used, particularly relating to use of only a single ice motion dataset and rejecting the use of satellite thickness observations. It would also be great to include more context for the calculated solid ice freshwater fluxes. Otherwise the manuscript seems to be in a good state and my remaining comments/edits are all quite minor.

**Dear Reviewer:**

We would like to thank you for the constructive comments to improve this manuscript. We agree that adding more satellite thickness observations will improve the estimations of sea ice fluxes and the local ice volume variations. Thus, we added satellite-based sea ice thickness observations (Landy et al., 2017) to improve our estimations as suggested. We also tried to add OSISAF drift to calculate the sea ice fluxes following comment #1 and editor's suggestion. However, because of its limited coverage and rough mesh grid in the bay, the usage of this data may cause some unquantifiable errors. And because the new version of NSIDC drift performs much better according to Hiroshi Sumata and Frank Kauker (pers. com.), we decided to use NSIDC drift only to calculate the sea ice fluxes. Because the sea ice volume variations are also estimated based on the satellite observations in the revised text, the title of our manuscript has been be modified as "Ensemble-based estimation of sea-ice volume variations in the Baffin Bay". Furthermore, we discussed the freshwater budget in the Baffin Bay and Labrador Sea in the discussion part (Section 4) of our revised manuscript and we compared our estimate of the freshwater volume stored in ice with estimate of previous studies. Additionally, we included maps of the monthly mean freshwater fluxes as well as the freshwater volume that is stored in the sea ice in the bay in our revised manuscript.

The specific responses and revisions are elucidated below shown in blue font for clarity.

**Corresponding Author: Qinghua Yang**

**Specific comments:**

**Point 1:** Three model ice volume products are used but only one drift product. Alternative drift vectors from OSISAF and/or Kimura et al could also be used to improve the determination of the volume flux uncertainty envelope. Line 62, is OSISAF not available year-round in BB? If other products are not available year-round or have full coverage over BB, can you estimate the uncertainty envelope for the ice motion for the seasons/region where they do overlap and use that in your determination of overall error?

Response 1: We agree with this constructive comment and decided added the wellvalidated low resolution OSISAF drift (OSI-405) to consolidate our estimates of seaice volume (SIV) fluxes in the bay. We chose the OSISAF drift rather than the KIMURA drift because the OSISAF and NSIDC drift data are proved to have valid performance in the Arctic (Sumata et al., 2014) and OSISAF drift data performs better than other data (i.e., NSIDC, CERSAT and KIMURA). However, for the study period, OSISAF drift data is only available for freezing season (October-April) while the summer season drift data is provided since 2017 (http://www.osi-saf.org/?q=content/sea-ice-products). Then we calculate the mean southward velocity over the northern inflow gate and southern outflow gate (Figure 1), and the SIV fluxes based on OSISAF drift were also calculated (not shown). The mean southward velocity from OSISAF is about 5.83 km d-1 over the north gate and 9.18 km d-1 over the south gate during the freezing season, respectively. However, the mean southward velocity based on NSIDC is only about 3.05 km d-1 and 3.69 km-1 during the freezing season, respectively. Results show that the mean southward and northward velocity from OSI-405 are much higher than NSIDC drift (V4) though the annual cycles of these two drift data are similar. This overestimation can be attributed to its (OSI-405) reduced spatial coverage in lower latitudes and rough mesh grid (62.5 km) in the bay (e.g., especially along the south gate in this study) which cause only a few sea ice floes with higher velocity located in the centre of the Davis Strait (i.e., south gate) can be measured. As previously discussed, it may induce some unquantifiable errors in Baffin Bay if OSISAF drift is solely used. The coverage of OSISAF and NSIDC drift in the freezing season of 2011 is shown in Figure 2 as an example.

Then, we have tried to average the OSISAF drift and the NSIDC drift during the freezing season to get an ensemble mean sea ice drift. And for the melting season (May-September), only NSIDC drift was used. Nevertheless, this method will also cause some bias because the systematic differences between NSIDC and OSI ice drift are found (as example in Figure 1). The inappropriateness of this method was also pointed out by the editor. Moreover, the new information that we got is the new version of NSIDC drift performs much better according to Hiroshi Sumata and Frank Kauker (*pers. com.*). So, based on above reasons, we decide to only use NSIDC drift to estimate the sea ice volume fluxes in the Baffin Bay.

All estimates of sea-ice volume (SIV) and of the fluxes have been recalculated with

NSIDC drift and the satellite-based sea ice thickness (will be discussed in Point 2) and the model ice thicknesses. The updated results are shown in in our revised manuscript. *(please see these updated results in our revised manuscript)*

Figure 1. Monthly mean southward velocity over the northern inflow gate (a) and southern outflow gate (b), respectively.

---

## Author Response (AR2)

**Response letter**

Dear Editor:

We thank you very much for the comments for the paper 'Ensemble-based estimation of sea-ice volume variations in the Baffin Bay' submitted to *the Cryosphere*. We have made a minor revision according to your comments. The specific texts are added to clarify the standard deviations: *And we use one standard deviation (i.e., +/-number) among these ensemble members (i.e., SIV fluxes estimated from different sea ice thickness data sets and NSIDC SID) to show the uncertainties of flux estimates in this study* (please see the red texts in line 151-153, Page 5).

Qinghua Yang

On behalf of all the authors